# The NFIB-ERO1A axis promotes breast cancer metastatic colonization of disseminated tumour cells

Federica Zilli[1,2,†], Pedro Marques Ramos[2,3,†], Priska Auf der Maur[1] (iD), Charly Jehanno[1] (iD), Atul Sethi[1,2,4], Marie-May Coissieux[1,2], Tobias Eichlisberger[2], Loïc Sauteur[1], Adelin Rouchon[1], Laura Bonapace[3], Joana Pinto Couto[1,2,3], Roland Rad[5,6], Michael Rugaard Jensen[3], Andrea Banfi[1] (iD), Michael B Stadler[2,4] & Mohamed Bentires-Alj[1,2,*] (iD)

## Abstract

**Metastasis is the main cause of deaths related to solid cancers. Active transcriptional programmes are known to regulate the metastatic cascade but the molecular determinants of metastatic colonization remain elusive. Using an inducible *piggyBac* (PB) transposon mutagenesis screen, we have shown that overexpression of the transcription factor nuclear factor IB (NFIB) alone is sufficient to enhance primary mammary tumour growth and lung metastatic colonization. Mechanistically and functionally, NFIB directly increases expression of the oxidoreductase *ERO1A*, which enhances HIF1α-VEGFA-mediated angiogenesis and colonization, the last and fatal step of the metastatic cascade. *NFIB* is thus clinically relevant: it is preferentially expressed in the poor-prognostic group of basal-like breast cancers, and high expression of the *NFIB/ERO1A/VEGFA* pathway correlates with reduced breast cancer patient survival.**

**Keywords** Breast cancer; ERO1A; metastasis; NFIB; VEGFA

**Subject Categories** Cancer; Signal Transduction; Vascular Biology & Angiogenesis

## Introduction

Breast cancer is the most frequently diagnosed cancer and the leading cause of death in female cancer patients worldwide (Fahad Ullah, 2019). Metastasis remains the primary cause of solid-cancer-evoked mortality (Gupta & Massagué, 2006). The multistep process of metastasis includes local tumour cell invasion, entry into the vasculature (intravasation), exit from the circulation into the parenchyma of distant organs (extravasation), and colonization. Colonization, which results in the development of clinically manifesting metastasis and fatal disease, is dependent upon resistance of the disseminated tumour cells (DTCs) to immune and host tissue defences, survival in a foreign environment and tumour initiation capacity (Pantel & Brakenhoff, 2004; Massagué & Obenauf, 2016). However, the molecular determinants of colonization remain elusive.

Numerous oncogenic mutations and other genomic alterations co-occur often in cancer cells and result in tumour heterogeneity, which impinges on the clinical outcome of the disease (Kennecke *et al*, 2010; Yates & Campbell, 2012; Koren & Bentires-Alj, 2015). The dynamic evolution of the cancer genome is influenced by the generation of additional mutations and selective forces acting on cancer clones (Kreso & Dick, 2014). Next-generation sequencing of DNA from clinical specimens has provided insights into breast cancer genetics and contributed to identifying potential drivers of tumour progression (Yates *et al*, 2015; Nik-Zainal *et al*, 2016; Pereira *et al*, 2016; Robinson *et al*, 2017; Angus *et al*, 2019; Bertucci *et al* 2019; De Mattos-Arruda *et al*, 2019). Mechanistic elucidation of the effects of co-occurring genomic alterations and their functional validation is required to define the causality between these events and their contribution to specific steps of tumour progression to overt metastases.

Activating mutations of *PIK3CA*, which encodes for the p110α catalytic subunit of phosphoinositide 3-kinase (PI3K), are among

1   Department of Biomedicine, Department of Surgery, University Hospital Basel, University of Basel, Basel, Switzerland
2   Friedrich Miescher Institute for Biomedical Research, Basel, Switzerland
3   Novartis Institutes for Biomedical Research, Basel, Switzerland
4   Swiss Institute of Bioinformatics, Basel, Switzerland
5   Department of Medicine II, TUM School of Medicine, Institute of Molecular Oncology and Functional Genomics, Center for Translational Cancer Research (TranslaTUM), Technische Universität München, München, Germany
6   German Cancer Consortium (DKTK), German Cancer Research Center (DKFZ), Heidelberg, Germany
    *Corresponding author. Tel: +41 61 26 53 313; E-mail: m.bentires-alj@unibas.ch
    †These authors contributed equally to this work

the most frequent alterations in human breast cancer and lead to an hyperactivated PI3K pathway signalling (Yuan & Cantley, 2008; Zhao & Vogt, 2008; Miller, 2012). We and others have shown that inducible expression of the *PIK3CA*[H1047R] mutation evokes heterogeneous mammary tumours in mice (Meyer *et al*, 2013; Koren & Bentires-Alj, 2013; Koren *et al*, 2015), which indicates a causative effect of *PIK3CA* mutations in mammary tumorigenesis. In contrast, metastases were not found in mice with *PIK3CA*[H1047R]-derived tumours (Koren & Bentires-Alj, 2013; Koren *et al*, 2015), which thus provides a model system to identify collaborating gain- or loss-of-function genomic alterations that contribute to metastasis.

Transposon insertional mutagenesis is a powerful tool in mice for discovering genes related to cancer (Ding *et al*, 2005; Dupuy *et al*, 2005; Rad *et al*, 2010; Dupuy, 2010). Indeed, the fact that transposons are mobile within the genome and alter gene activity in cells that express the transposase makes these systems ideal for whole-genome screens. The *piggyBac* (PB) transposon was engineered to be active in mammalian cells (Ding *et al*, 2005) and it allows functional identification not only of oncogenes but also of tumour suppressor genes, depending on the site of insertion and the orientation of the transposon (Rad *et al*, 2015; Takeda *et al*, 2017; De La Rosa *et al*, 2017; Weber *et al*, 2019).

We performed an unbiased PB transposon insertional mutagenesis screen in cancer cells with an activating *PIK3CA*[H1047R] mutation to identify possible synergistic mechanisms of metastatic colonization. Mechanistically and functionally, we demonstrate that NFIB enhances the expression of the oxidoreductase *ERO1A* and *VEGFA*, promotes metastatic colonization and shortens overall survival of mice.

# Results

## Transposon mutagenesis confers metastatic potential to *PIK3CA*[H1047R] mammary cells

To identify cancer genes relevant to metastatic progression, we engineered a murine non-metastatic mammary cancer cell line derived from *PIK3CA*[H1047R] mutant tumours (LB-mHR1, here cited as HR1) with a doxycycline (dox)-inducible *piggyBac* (PB) transposon system (HR1. PB) (Ivics *et al*, 2009; Rad *et al*, 2010) (Fig 1A). HR1 cells lacking the transposon served as control (HR1. Ctrl). To perform an unbiased *in vivo* PB transposon mutagenesis screen (Fig 1B), we injected HR1. PB and HR1. Ctrl orthotopically into 19 and 14 NOD/SCID mice, respectively, and monitored the animals for tumour growth and metastasis. PB transposon mutagenesis in HR1. PB cells increased tumour incidence (Fig EV1A) and promoted metastasis (Fig EV1B). Metastases were observed in half of the mice injected with HR1. PB cells of which nine lung metastases (LM)-derived cell lines were isolated. To phenotypically characterize the LM cell lines *in vivo*, we injected them into mammary fat pads of NOD/SCID mice. The LM1, LM8 and LM9 lines exhibited a particularly aggressive behaviour characterized by accelerated tumour growth and metastasis (Fig EV1C–G). These results indicate that selected transposon integrations can confer aggressive metastatic behaviour on cells.

## Insertional PB-mutagenesis screening identifies *Nfib* as a metastatic gene in mammary cancer

To investigate induced alterations that enabled cancer cells to metastasize, we mapped transposon integration sites in 16 primary tumours and 18 metastatic samples, searching for insertions enriched at metastatic sites compared with tumours. We used splinkerette PCR followed by next-generation sequencing from both transposon arms (Friedrich *et al*, 2017) and identified 1,252 insertions in 1,080 genes (Dataset EV1). To evaluate the relative abundance of each integration site and minimize PCR-induced amplification effects, we devised a diversity count measure for each integration site within each sample (percentage of unique insertions) that quantifies the number of unique sheared ends rather than all sequencing counts. To select metastatic genes, we searched for genes with more unique insertion sites (based on their diversity counts) in metastatic samples than in tumour samples (Fig 1C). Depending on the transposon integration site, insertions may lead to gene activation or inactivation. The most frequently altered genes in the primary tumours and/or metastatic samples were *Lrp1b*, *Nfia*, *Foxp1*, *Nfib*, *Lrch3* and *Sox6*. Some of these unique insertions (e.g. *Foxp1*) were found in both metastatic and non-metastatic samples at high frequency, suggesting that they are unlikely to be specifically required for metastasis. Notably, unique insertions in *Nfib* were particularly enriched in a subset of highly metastatic samples (LM1/8/9) compared with the other hits (Fig 1D). The same-strand insertions upstream of both *Nfib* and *Foxp1* transcription start sites suggest them to be candidate oncogenes. Consistent with the pattern of transposon integration, analysis of *Nfib*/*Foxp1* mRNA expression and NFIB/FOXP1 protein abundance in the LM cell lines revealed enhanced levels of FOXP1 in all LM cell lines but selective elevation of NFIB in LM lines with high metastatic potential (Fig 1E–G). These data suggest the importance of *Nfib* in mammary cancer metastasis.

## Depletion of *Nfib* delays mammary cancer growth and abrogates metastases

To assess the contribution of *Nfib* and *Foxp1* to tumour growth and metastatic progression, we produced knockout (KO) cells for the two genes using CRISPR-Cas 9 technology in the highly metastatic LM1 and LM9 lines and generated oligoclonal pools of cells (Appendix Fig S1A and B). Considering that NFIB has been implicated in skin stem cell maintenance (Chang *et al*, 2013) and tumour growth in other cancer types (Brayer *et al*, 2016; Semenova *et al*, 2016; Wu *et al*, 2016; Fane *et al*, 2017; Wu *et al*, 2018), we addressed its effects on mammary tumours. *Nfib* KO in LM1 and LM9 delayed tumour growth (Fig 2A) and markedly decreased the frequency of tumour-initiating cells (TICs) (Fig EV2A and B). Consistently, *Nfib* KO also reduced tumoursphere formation (Fig EV2C and D).

Deletion of *Foxp1* increased mammary tumour latency (Appendix Fig S1C) and slightly decreased metastasis compared to the wild type (WT) (Appendix Fig S1D). In contrast, when LM1 and LM9 *Nfib* KO cells were injected orthotopically and the primary tumour removed, we observed a dramatic abrogation of lung metastasis (orthotopic metastasis assay) compared with controls (Fig 2B

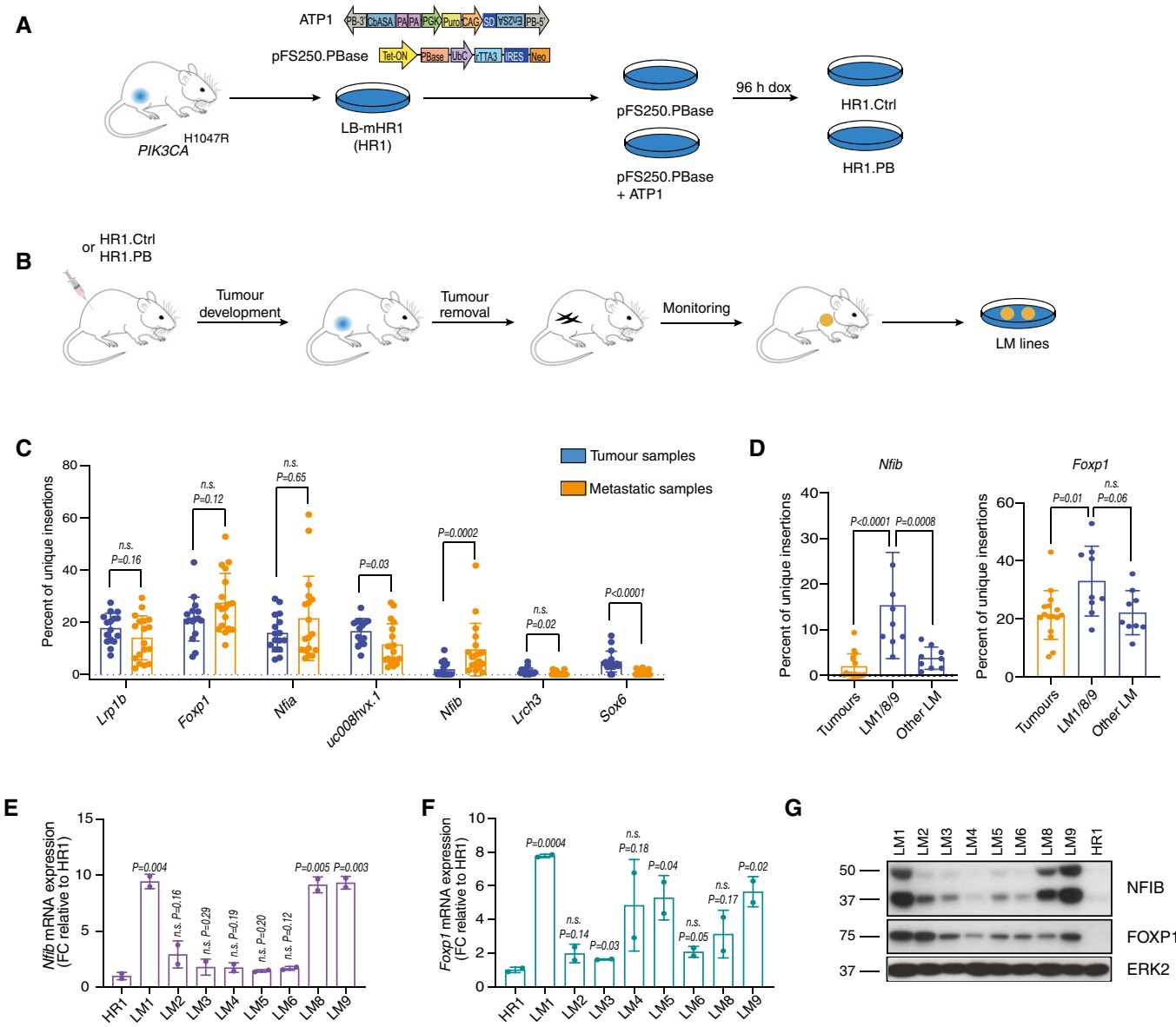

**Figure 1.  Transposon mutagenesis screen identifies *Nfib* as a candidate metastatic inducer in breast cancer.**

A   Screen design: generation of mammary cancer cell lines LB-mHR1 (HR1) from *PIK3CA*[H1O47R] mutant transgenic animals and transfection with the *piggyBac* (PB) system (HR1 Ctrl and HR1. PB cells) (Ivics *et al*, 2009). PB transposon (ATP1-Puro) and transposase (pFS250. PBase) plasmid design. PB-3′/5′, *piggyBac* inverted terminal repeats; CβASA, Carp β-actin splice acceptor; En2SA, Engrailed-2 exon-2 splice acceptor; SD, Foxf2 exon-1 splice donor; pA, bidirectional SV40 polyadenylation signal; PGK, mouse phosphoglycerate kinase 1 promoter; Puro, puromycin resistance; CAG, cytomegalovirus enhancer and chicken beta-actin promoter; Tet-ON, tetracycline-responsive element-tight promoter; PBase, PB transposase; UbC, Ubiquitin C promoter; rTTA3, reverse tetracycline transactivator 3; IRES, internal ribosomal entry site; Neo, Neomycin/G418 selection marker. The cell pools were treated with doxycycline (1 μg/ml) for 96 h *in vitro*.

B   *In vivo* screen design: generation of lung metastatic mammary cancer cell lines (LM) from HR1. PB cell lines after orthotopic injections of HR1. PB or HR1. Ctrl.

C   Bar graph showing the percentage of unique insertions in the top 7 genes normalized to the total number of insertions in a given sample. Genes are annotated with a gene symbol when available otherwise with UCSC ID transcript names. Data are from all the sequenced samples (tumour samples: 16 primary tumours; metastatic samples: 3 lung micro-metastases, 6 lung macro-metastases and 9 LM cell lines). Means ± s.d., two-tailed Mann–Whitney *U*-test, n.s. = not significant.

D   Bar graphs showing the percentage of unique insertions in *Nfib* and *Foxp1* in tumours, LM1, LM8, LM9 and other LM cell lines normalized to the total number of insertions in a given sample. Dots represent individual samples (Tumours *n* = 16, LM1/8/9 *n* = 9, Other LM *n* = 9), means ± s.d., two-tailed Mann–Whitney *U*-test, n.s. = not significant.

E   Bar graph representing mean *Nfib* mRNA expression normalized to HR1 cells. *n* = 2 biological replicates and *n* = 2 technical replicates, means ± s.d., two-tailed Student's *t*-test, FC = fold change.

F   Bar graph representing mean *Foxp1* mRNA expression normalized to HR1 cells. *n* = 2 biological replicates and *n* = 2 technical replicates, means ± s.d., two-tailed Student's *t*-test, FC = fold change.

G   Immunoblot analysis of LM and HR1 cell lines showing NFIB and FOXP1 protein levels. ERK2 served as a loading control.

Source data are available online for this figure.

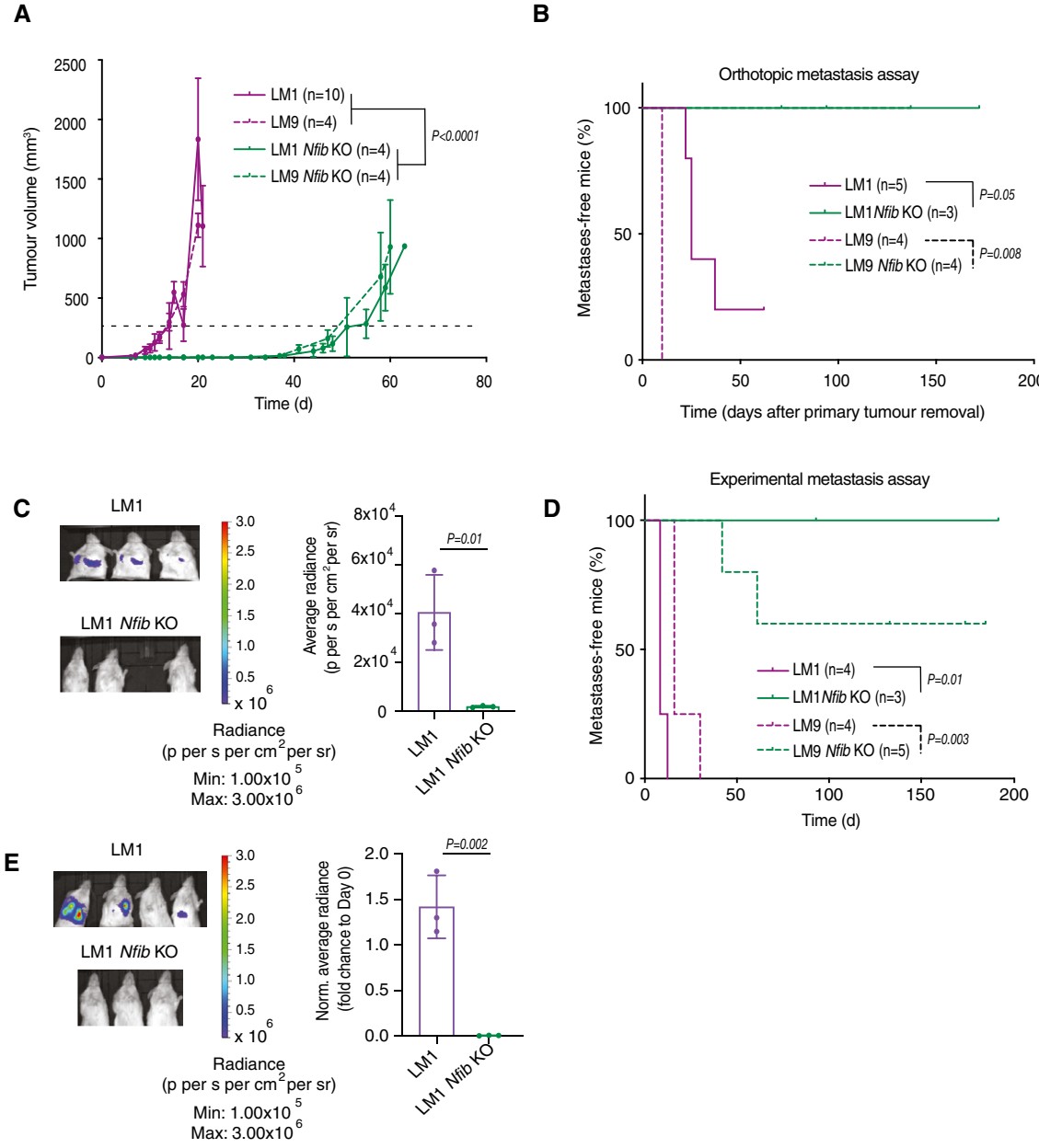

**Figure 2. *Nfib* ablation delays mammary tumour formation and abrogates metastasis.**

A   The kinetics of LM1 (*n* = 10), LM9 (*n* = 4), LM1 *Nfib* KO (*n* = 4), and LM9 *Nfib* KO (*n* = 4) tumour growth upon orthotopic injection of 250 × 10³ cells into NOD/SCID mice. Curves show means of tumour volume ± s.d., two-way ANOVA group analysis of the times to reach 250 mm³ (dashed line).

B   Kaplan–Meier plot depicting metastatic onset after tumour removal in mice injected orthotopically with LM1 (*n* = 5), LM9 (*n* = 4), LM1 *Nfib* KO (*n* = 3) or LM9 *Nfib* KO (*n* = 4) cells, two-tailed log-rank test.

C   *Nfib* knockout abrogates metastasis in the orthotopic LM1 model. Representative bioluminescence images (left panel) and bar plot quantification (right panel) of lung metastases at 25 (LM1) and 40 (KO) days after primary tumour removal; *n* = 3 mice, means ± s.d., two-tailed Student's *t*-test.

D   *Nfib* knockout impairs experimental metastases formation in the LM models. Kaplan–Meier plot showing metastatic incidence of animals inoculated *i.v.* with LM1 (*n* = 4), LM9 (*n* = 4), LM1 *Nfib* KO (*n* = 3), or LM9 *Nfib* KO (*n* = 5) cells, two-tailed log-rank test.

E   Representative bioluminescence images (left panel) and bar plot quantification (right panel) of lung metastases 16 days after *i.v.* injection of LM1 or LM1 *Nfib* KO cells, *n* = 3 mice, means ± s.d., two-tailed Student's *t*-test.

Source data are available online for this figure.

and C), providing a rationale to focus on the effect of *Nfib* in breast cancer metastasis. The data suggest that the *Nfib* overexpression observed in our PB screen and in metastatic lines may also enhance

metastatic colonization. To test this possibility, we injected *Nfib* KO or control cells intravenously (*i.v.*, experimental metastasis assay) and found that *Nfib* KO completely or dramatically impaired

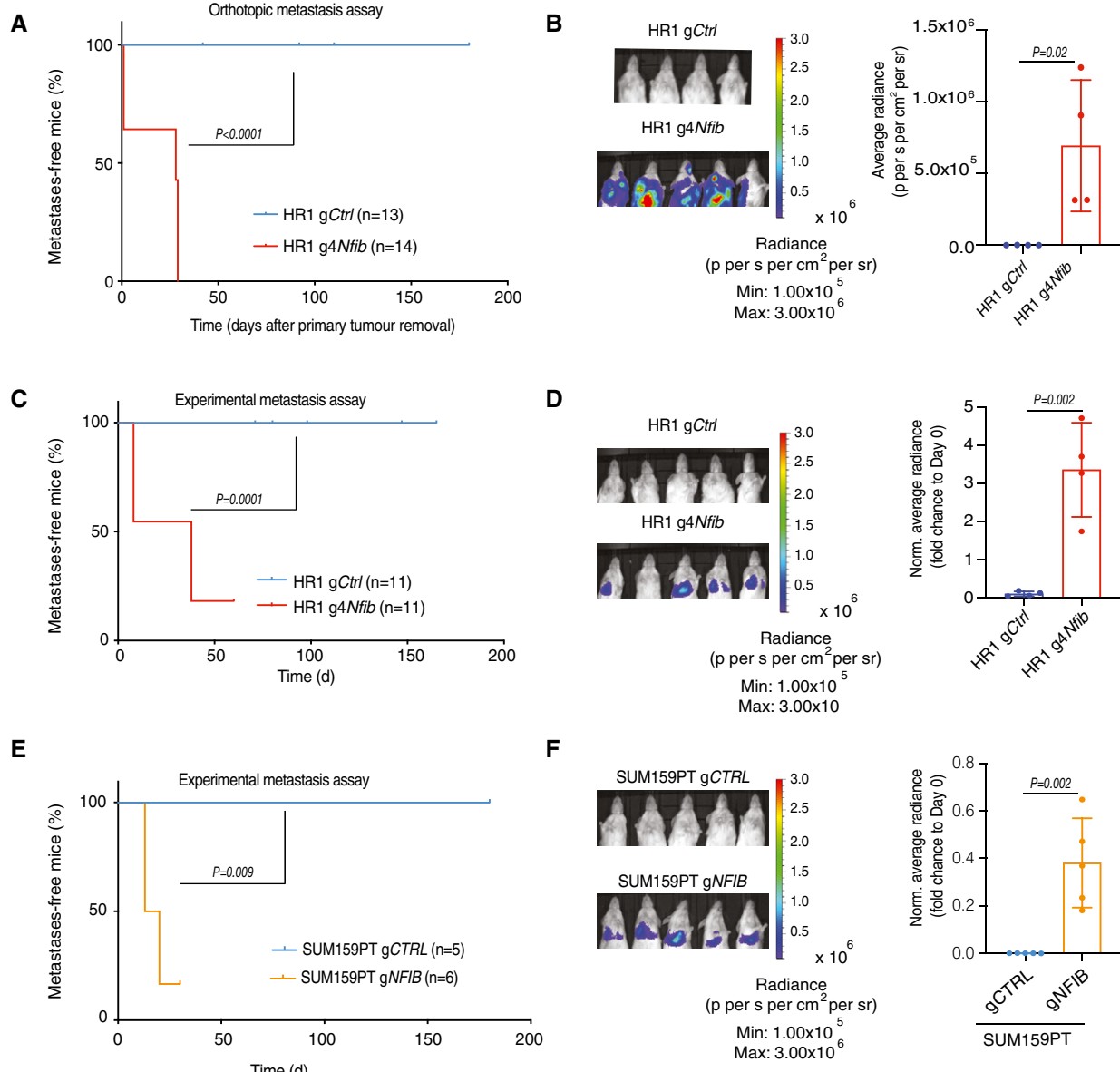

**Figure 3. *Nfib* is sufficient to induce metastasis.**

A  Kaplan–Meier plot depicting metastasis onset after tumour removal in mice injected orthotopically with HR1 g*Ctrl* (*n* = 13) or HR1 g4*Nfib* (*n* = 14), two-tailed log-rank test.

B  *Nfib* overexpression in HR1 parental cells enhances metastasis in the orthotopic model. Representative bioluminescence images (left panel) and bar plot quantification (right panel) of lung metastases at day 20 (HR1 g*Ctrl*) and day 2 (HR1 g4*Nfib*) after primary tumour removal. *n* = 4 mice, means ± s.d., two-tailed Student's *t*-test.

C  *Nfib* overexpression in the HR1 parental cells enhances experimental metastases. Kaplan–Meier showing metastatic incidence of animals inoculated *i.v.* with HR1 g*Ctrl* (*n* = 11) or HR1 g4*Nfib* (*n* = 11) cells, two-tailed log-rank test.

D  Representative bioluminescence images (left panel) and bar plot quantification (right panel) of lung metastases 16 days after *i.v.* injection of HR1 g*Ctrl* or HR1 g4*Nfib* cells. *n* = 4, means ± s.d., two tailed Student's *t*-test.

E  *NFIB* overexpression enhances experimental metastases formation in the SUM159PT model. Kaplan–Meier survival analysis of animals inoculated *i.v.* with SUM159PT g*CTRL* (n = 5) or SUM159PT g*NFIB* (n = 6) cells, two-tailed log-rank test.

F  Representative bioluminescence images (left panel) and bar plot quantification (right panel) of lung metastases 20 days after *i.v.* injection of SUM159PT g*CTRL* or SUM159PT g*NFIB* (n = 5), means ± s.d., two-tailed Student's *t*-test.

Source data are available online for this figure.

metastatic colonization in the LM1 and LM9 lines, respectively (Fig 2D and E). Finally, ablation of *Nfib* using CRISPR-Cas 9 technology also increased tumour latency, decreased the number of circulating tumour cells and prolonged overall survival of BALB/c mice orthotopically injected with the highly metastatic 4T1 mammary cancer cells (Appendix Fig 2A–D). The results of these functional assays demonstrate the importance of NFIB in mammary cancer metastatic colonization.

## NFIB is sufficient to induce metastasis

We next asked whether NFIB activity is alone sufficient for inducing metastasis. Using Cas9-Activators with Synergistic Activation Mediators (SAM) (Konermann *et al*, 2015), we overexpressed *Nfib* from its endogenous promoter in the non-metastatic parental HR1 cell line (Fig EV3A). Two different guides were designed and tested for their ability to increase *Nfib* expression and g4*Nfib* was selected for our experiments. *Nfib* overexpression enhanced proliferation, tumoursphere formation, migration and invasion *in vitro* (Appendix Fig S3A–D). Furthermore, overexpression of *Nfib* accelerated tumour onset (Fig EV3B), metastasis formation after orthotopic injection (Fig 3A and B) and metastatic colonization after tail-vein injection (Fig 3C and D). Overexpression of *NFIB* in SUM159PT, a human TNBC cell line with low metastatic potential (Fig EV3C), similarly decreased tumour latency (Fig EV3D) and increased metastasis, both in the orthotopic (Fig EV3E and F) and the experimental metastasis assays (Fig 3E and F). Thus, NFIB appears to be sufficient to induce metastasis when overexpressed in non- (HR1) and low- (SUM159PT) metastatic mammary cancer lines.

## NFIB induces metastatic colonization via increased *Ero1l/ERO1A* expression

To determine the molecular mechanism underlying increased metastatic colonization driven by NFIB, we sequenced total mRNA of tumourspheres and primary tumours derived from *Nfib* high-expression models (LM1 and LM9) and *Nfib* low-expression models (HR1, LM1 *Nfib* KO and LM9 *Nfib* KO). By comparing the 200 most differentially expressed genes with ChIP-seq data of putative transcriptional targets of *Nfib* in the mammary gland

(Shin *et al*, 2016) and epithelial-melanocyte stem cells (Chang *et al* 2013; Dataset EV2–EV4), we found endoplasmic reticulum disulphide oxidase 1 like (*Ero1l*) to be the single common gene (Fig 4A and B). Chromatin immunoprecipitation (ChIP) qPCR for NFIB in the LM1/LM1 *Nfib*KO, LM9/LM9 *Nfib*KO, 4T1 WT, 4T1 *Nfib* KO, HR1gCtrl and HR1g4*Nfib* cell lines, revealed increased NFIB binding to the promoter of *Ero1l* (Shin *et al*, 2016) (Fig 4C), indicating that ERO1L is a direct transcriptional target of NFIB. Indeed, abundance of the mouse and human ERO1L/A protein was increased in *Nfib/NFIB*-overexpressing cells (Appendix Fig S4A). Consistently, *Nfib* KO decreased *Ero1l* mRNA levels (Appendix Fig S4B–D). ERO1L/ERO1A is involved in the production of hydrogen peroxidase ($H_2O_2$) (Zito, 2015) and is a poor-prognostic factor in cancer (Takei *et al*, 2017; Zhou *et al*, 2017; Kim *et al*, 2018; Yang *et al*, 2018).

We next examined whether *Ero1l/ERO1A* mediates NFIB effects on metastatic colonization. Notably, knockdown of *ERO1A* in SUM159PT g*NFIB*, LM1 and HR1 g4*Nfib*, using two doxycycline-inducible shRNA constructs (Appendix Fig S5A) reduced metastatic colonization after tail-vein injection and prolonged survival (Fig 4D–F and Appendix Fig S5B–E). Altogether, these findings show that *Ero1l/ERO1A* expression is increased in *Nfib/NFIB*-overexpressing models and suggest that the NFIB-ERO1A axis is critical for metastatic colonization in breast cancer.

## ERO1A induces VEGFA expression and angiogenesis via HIF1α stabilization

We found that *NFIB/Nfib*-ERO1A/*Ero1l* overexpressing cells produce more ROS compared to the low-expression models (Fig 5A). ROS has been shown to stabilize HIF1α (Bell *et al*, 2007; Yan *et al*, 2010), and we found an increase in nuclear HIF1α in the *NFIB/Nfib*-ERO1A/*Ero1l* overexpressing models which was offset by *ERO1A* knockdown (Fig 5B). As ERO1A and HIF1α have been shown to enhance *VEGFA* expression (Amelino-Camelia *et al*, 1998; Tanaka *et al*, 2016), we assessed *VEGFA* mRNA and protein levels in *NFIB* models. VEGFA increased in cells, mammary tumours and lung metastases derived from *NFIB/Nfib*-overexpressing models and decreased upon *ERO1A* downregulation (Fig 5C and EV4A–D, Appendix Fig S6A–C). Downregulation of *Nfib* in HR1 g4*Nfib* cells decreased

---

**Figure 4. The NFIB-ERO1A axis enhances lung metastatic colonization.**

A, B   Intersection of the top 200 differentially expressed genes (sorted by statistical significance) in tumourspheres and tumours (*Nfib* high-expression vs. the respective controls) and putative transcriptional targets of *Nfib* determined by ChIP-seq in mammary gland (Shin *et al*, 2016) (A) and epithelial-melanocyte stem cells (Chang *et al*, 2013) (B) (association rule: Basal + extension: 5,000 bp upstream, 1,000 bp downstream, 10,000 bp max extension, curated regulatory domains included).

C   ChIP was performed in all murine cell lines (*Nfib* high-expression and the respective *Nfib* low-expression or KO controls) against *Nfib* followed by qPCR with primers specific for *Ero1l* promoter region proximal to transcription starting site (Ero1l_492 and Ero1l_493). Meg3 was used as control. Graph shows qPCR results with % input method as indicated (*Nfib* high-expression vs. the respective *Nfib* low-expression or KO controls). *n* = 2 biological replicates and *n* = 2 technical replicates, means ± s.d., two-tailed Student's *t*-test, *n.s.* = not significant.

D   Kaplan–Meier survival analysis of NSG mice inoculated *i.v.* with SUM159PT g*NFIB* sh*CTRL* (*n* = 9), SUM159PT g*NFIB* sh1 *ERO1A* (*n* = 7), or SUM159PT g*NFIB* sh2 *ERO1A* (*n* = 7) cells. Two-tailed log-rank test.

E   Representative bioluminescence images (left panel) and bar plot quantification (right panel) of lung metastases 8 (SUM159PT g*NFIB* sh*CTRL*) and 20 (SUM159PT g*NFIB* sh1 *ERO1A* or SUM159PT g*NFIB* sh2 *ERO1A*) days after *i.v.* injection of the respective cells. *n* = 4, means ± s.d., two-tailed Student's *t*-test.

F   Kaplan–Meier survival analysis of overall survival of mice injected *i.v.* with SUM159PT g*NFIB* sh*CTRL* (*n* = 10), SUM159PT g*NFIB* sh1 *ERO1A* (*n* = 12), or SUM159PT g*NFIB* sh2 *ERO1A* (*n* = 13) cells, two-tailed log-rank test.

Source data are available online for this figure.

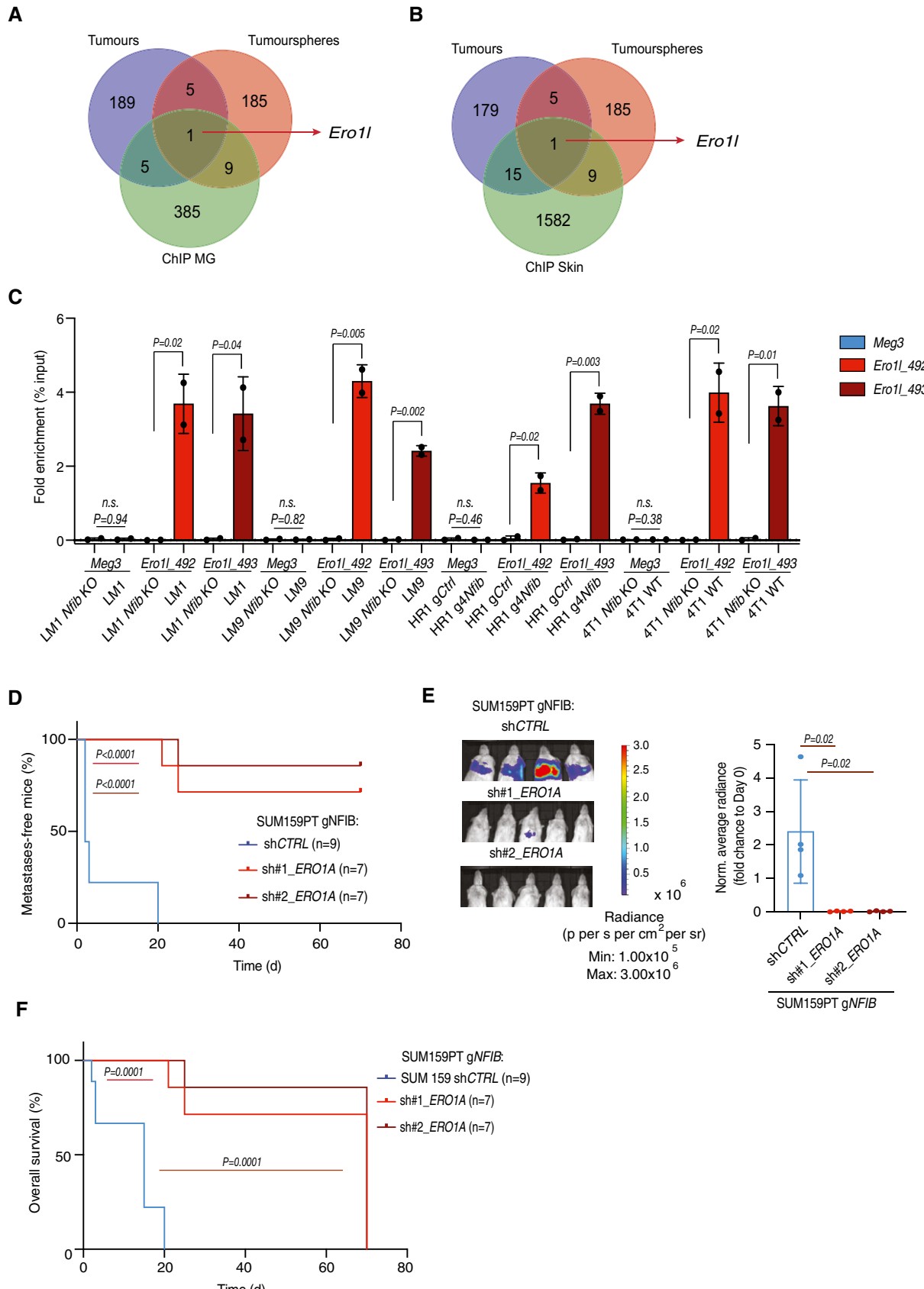

Figure 4.

*Vegfa* and VEGFA levels (Appendix Fig S6D–F). Consistently, rescued expression of *Ero1l* in LM1, LM9 and 4T1 *Nfib* KO cell lines restored *Vegfa* mRNA and VEGFA protein levels and increased metastatic colonization after tail-vein injection (Fig EV5A–E), indicating that *VEGFA* expression is *NFIB/ERO1A* dependent. Together, the data suggest that the NFIB-ERO1A axis enhances ROS-evoked HIF1α stabilization and VEGFA levels.

We assessed angiogenesis by endothelial-specific CD31 staining in sections of mouse mammary tumours and metastases from NFIB overexpression and KO models and observed higher and lower frequency of vascular structures than controls, respectively (Fig EV4E-G). CD31-positive blood vessels were also decreased in lung metastases upon *ERO1A* knockdown in SUM159PT gNFIB cells (Fig 5D and Fig EV4G). *In vitro*, conditioned medium from *NFIB/Nfib/Ero1l/ERO1A* high-expression cell

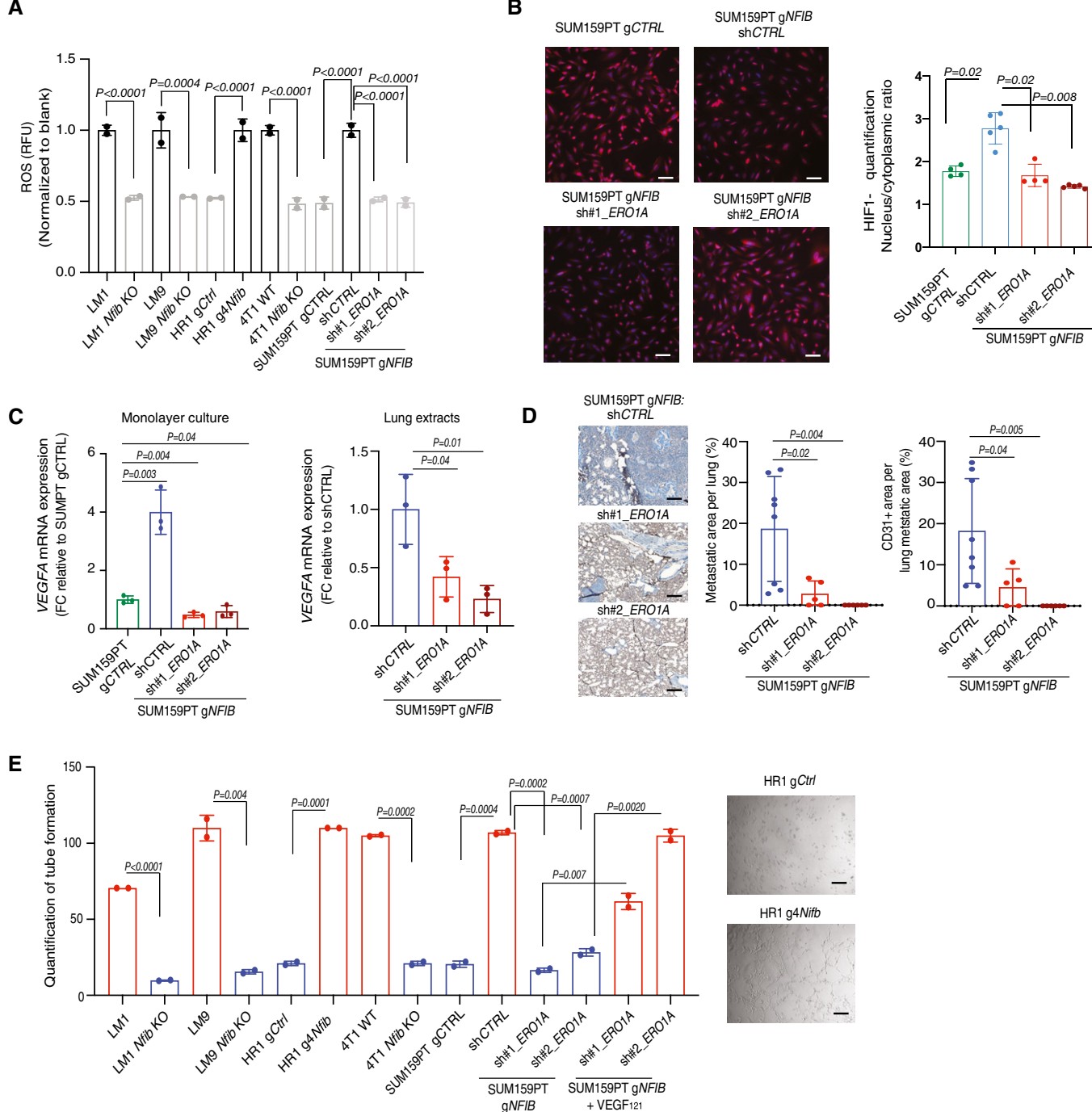

**Figure 5.**

**Figure 5. *Ero1l/ERO1A* increases *Vegfa/VEGFA* expression and angiogenesis.**

A Bar graph showing the ROS levels in the NFIB models. The fluorescence signal was normalized to the blank. $n = 2$ biological replicates and $n = 2$ technical replicates, means $\pm$ s.d., two-tailed Student's *t*-test.

B Left panel: representative images of HIF1α immunofluorescence. HIF1α was stained with Alexa-633 (red) and nucleus with DAPI (blue). Scale bar, 100 μm. Right panel: bar graph showing quantification of HIF1α nuclear/cytoplasmatic ratio. $n = 5$ biological replicates, means $\pm$ s.d., two-tailed Mann–Whitney *U*-test.

C Left panel: Bar graph representing mean *VEGFA* mRNA expression in SUM159PT gCTRL and SUM159PT gNFIB breast cancer cell lines upon downregulation of *ERO1A* with two independent shRNAs (sh1 and sh2). Right panel: Bar graph representing mean *VEGFA* mRNA expression in lung extracts from animals inoculated *i.v.* with SUM159PT gNFIB shCTRL, SUM159PT sh1 *ERO1A*, or SUM159PT sh2 *ERO1A* cells. In both graphs $n = 3$ biological replicates and $n = 2$ technical replicates, means $\pm$ s.d., two-tailed Student's *t*-test, FC = fold change.

D Left panel: Representative images of CD31-positive endothelial structures in lungs. Scale bar, 1 mm. Right panel: Bar graphs showing lung metastases quantified as percentage of metastatic area per lung area and quantification of CD31 staining in the metastatic area. Means $\pm$ s.d., $n = 8$ shCTRL, $n = 5$ per sh1 *ERO1A*, and $n = 6$ sh2 *ERO1A*, two-tailed Student's *t*-test.

E Left panel: Bar graph showing average of master segments counted after incubation of HUVEC cells with conditioned medium from LM1, HR1, 4T1 (*Nfib* high-expression and the respective *Nfib* low-expression or KO controls) and SUM159PT gNFIB shCTRL, SUM159PT gNFIB sh1 *ERO1A*, SUM159PT gNFIB sh2 *ERO1A* cells. Human VEGF$_{121}$ (0.4 ng/μl) was added to SUM159PT gNFIB sh1 and sh2 *ERO1A* cells, $n = 2$ technical replicates, means $\pm$ s.d., two-tailed Student's *t*-test. Right panel: Representative images showing tube formation four hours after addition of conditioned medium from HR1 gCtrl or g4Nfib. Scale bar, 100 μm.

Source data are available online for this figure.

lines increased endothelial tube formation compared with low-expression models in the matrigel-based tube formation assay. Addition of human recombinant VEGF$_{121}$ to the conditioned medium from *ERO1A* low-expression models enhanced endothelial tube formation (Fig 5E). Taken together, these data indicate that NFIB is a breast cancer metastasis transcriptional regulator which, through increased expression of *ERO1A* and *VEGFA*, promotes angiogenesis at the metastatic site. This creates a permissive microenvironment for metastatic colonization.

**High *NFIB-ERO1A-VEGFA* expression is associated with poor prognosis in TNBC patients**

*NFIB* is expressed preferentially in basal-like breast cancers and is a potential prognostic factor in TNBC human breast cancer, as observed in previous reports (Moon *et al,* 2011; Liu *et al,* 2019). NFIB overexpression discriminated between metastatic *versus* non-metastatic TNBC breast cancer patient-derived xenograft (PDX) models (Fig 6A; Appendix Fig S7A) and NFIB, ERO1A and VEGFA were co-overexpressed in these samples (Fig 6A and B). Next, analysis of *NFIB* mRNA expression levels from METABRIC (Fig 6C) (Curtis *et al,* 2012) and TCGA (Koboldt *et al,* 2012) (Appendix Fig S7B) confirmed that *NFIB* expression is elevated in basal-like breast cancer, according to PAM50 classification, and in iC10 by integrated cluster classification (Dawson *et al,* 2013; Mukherjee *et al* 2018). Increased NFIB gene copy numbers were observed in TNBC (Appendix Fig S7C and D). Notably, we found that high expression of *NFIB* and the combined signature of *ERO1A* and *NFIB* in patients with breast cancer of the basal-like subtype were predictive of decreased distant metastasis-free survival and/or overall survival (Fig 6D and E; Appendix Fig S7E and F).

# Discussion

Metastasis, the final, lethal hallmark of cancer, remains a major burden for breast cancer patients but there are numerous mechanisms that may prevent circulating cancer cells from colonizing distant organs. Only when these are circumvented and metastases develop is the prognosis of patients dismal (Massagué & Obenauf,

2016). Several interacting oncogenic pathways contribute to metastasis, but the exact molecular mechanisms of colonization are not fully understood. A detailed understanding of these molecular mechanisms is urgently needed.

Previous studies showed that *NFIB* governs epithelial-melanocyte stem cell behaviour and facilitates melanoma cell migration and invasion (Chang *et al,* 2013; Fane *et al,* 2017). NFIB has also been implicated in models of small-cell lung carcinoma (SCLC) and breast cancer (Moon *et al,* 2011; Denny *et al,* 2016; Semenova *et al,* 2016; Campbell *et al,* 2018; Liu *et al,* 2019), and *NFIB* or *ERO1A* has been associated with different tumour types (Becker-Santos *et al,* 2017; Zhou *et al,* 2017; Yang *et al,* 2018). Furthermore, in accordance with our results, ERO1A has been shown to enhance *VEGFA* expression in a TNBC human cell line (Tanaka *et al,* 2016). NFIB was shown to enhance TNBC cell survival and progression by suppressing *CDKN1A* (Liu *et al,* 2019), and to confer oestrogen independency in oestrogen receptor-positive breast cancer models (Campbell *et al,* 2018).

Here, we provide new insights into the effects of the NFIB-ERO1A-VEGFA axis on breast cancer progression to metastasis. Via an unbiased *in vivo* PB functional genetic screen, we have shown that NFIB induces mammary cancer metastatic progression and colonization. We have collected functional and mechanistic evidence that NFIB is a mammary cancer metastatic transcriptional regulator. First, depletion of NFIB delayed mammary tumour growth and abrogated metastases in three different cell lines when using both the orthotopic and experimental metastasis assays. Second, *Nfib* KO reduced tumoursphere formation and the frequency of tumour-initiating cells. Third, endogenous Nfib/NFIB overexpression was sufficient to evoke metastasis in non-metastatic murine and human mammary cancer lines in both the orthotopic and experimental metastasis assays. It also decreased overall survival of the animals.

Mechanistically, we have shown that ERO1A is a direct target of NFIB and a critical effector in evoking metastatic colonization. ERO1A enhanced ROS levels, causing HIF1α stabilization, increased VEGFA levels and angiogenesis, which resulted in a permissive microenvironment for colonization. These recorded effects of the NFIB-ERO1A-VEGFA axis on breast cancer progression to metastasis identify a targetable network for cancer therapy.

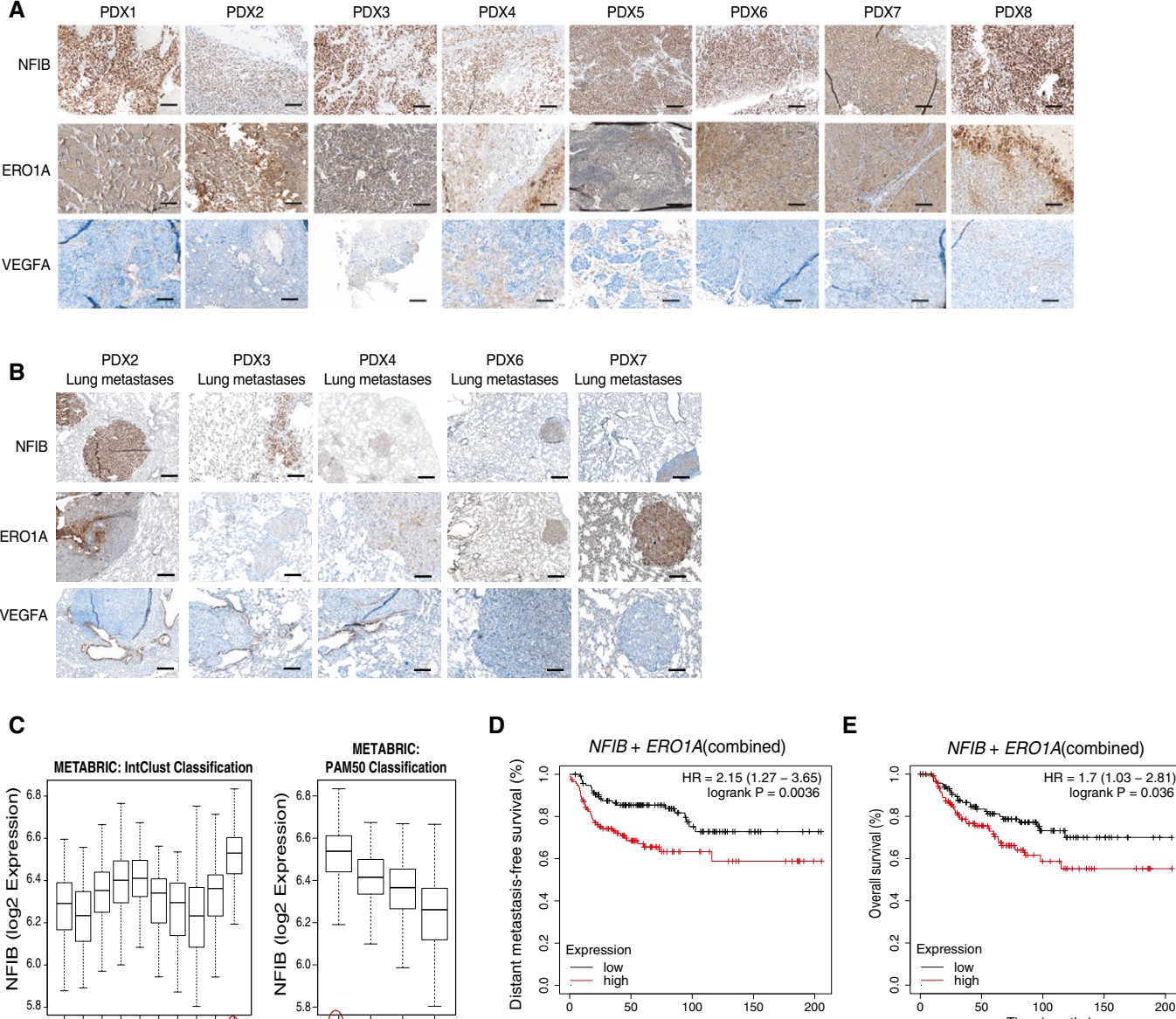

**Figure 6. NFIB-ERO1A-VEGFA overexpression is associated with aggressive tumours.**

A    Representative images of NFIB, ERO1A, VEGFA-positive TNBC PDX primary tumours. Scale bar, 1 mm.

B    Representative images of NFIB, ERO1A, VEGFA-positive lung metastases from TNBC PDX. Scale bar, 1 mm.

C    Higher *NFIB* expression in iC10 and basal breast cancer subtype compared with the other subtypes using intClust (Dawson *et al*, 2013) and PAM50 classifications in the METABRIC cohort (*n* = 1,980 across the 10 Intclust, iC). Boxplot represents *NFIB* expression values comprised between the 1st and the 3rd quartiles with the central band representing median expression value, and whiskers indicating the farthest data points that are still within the distance of 1.5 times the interquartile range (IQR).

D, E    Distant metastasis-free survival (D, *n* = 232) and overall survival (E, *n* = 241) plots generated using the Kaplan–Meier plotter based on the mean expression using the signal intensity of the *NFIB* (213029_at) combined with the *ERO1A* (218498_s_at) probes. The cut-off was automatically set to split patients into two groups (median), high and low. The plots were generated using the signal intensity of the different probes in Affymetrix microarray gene expression data from TNBC patients of The Cancer Genome Atlas. Number of patients (*n*) and *P* values (two tailed log-rank test) are presented in the panels.

Source data are available online for this figure.

# Materials and Methods

### Cell lines

LB-mHR1 is a mouse mammary cancer cell line derived from a transgenic mouse expressing $PIK3CA^{H1047R}$ under the CAG promoter (Meyer *et al* 2011). In brief, a mammary tumour was dissociated into single cells through mechanical and enzymatic digestion. The cells were sorted using GFP and established in DMEM supplemented with 10% FBS. To generate HR1. PB cells, LB-mHR1 cells were engineered with the PB system (Ivics *et al*, 2009; Rad *et al*, 2010) by transduction with the pFS250-PBase vector (HR1. Ctrl) followed by transfection with an ATP1 construct. To induce transposon mobilization and generate pools of mutagenized cells, HR1. PB cells were treated with dox for up to 96 h *in vitro*. LM (1 to 9) cell lines are lung-derived tumour cell lines isolated from NOD/SCID mice bearing HR1. PB-derived mammary tumours. In brief, lungs were dissociated into single cells and cultured until tumour cell colonies became apparent. For increased purity, tumour cells were sorted for GFP. HEK293T and 4T1 cell lines were purchased from the ATCC and cultured according to the ATCC protocol. LB-mHR1, 4T1, LM1-9 and HEK293T cells lines were cultured in DMEM supplemented with 10% FCS. SUM159PT cells were kindly provided by Dr. Charlotte Kupperwasser (Boston, USA). SUM159PT cells were cultured in Ham's F12 with 5% foetal calf serum (Gibco, Invitrogen), 5 μg/ml bovine insulin (Sigma), 1 μg/ml hydrocortisone (Sigma) and 1× penicillin/streptomycin (Gibco, Invitrogen). SUM159PT cell line identity was confirmed and routinely tested using short tandem repeat (STR) sequencing. HUVEC cells were cultured in endothelial growth medium 2 (EGM2 basal medium, Promocell) completed with endothelial growth supplements according to manufacturer's instructions and 1% penicillin/streptomycin. HUVEC were used at passage 5–6. All cell lines were tested routinely for mycoplasma contamination.

### PDX models

The PDXs used in this study were previously described (Derose *et al*, 2011; Gao *et al*, 2015) and their metastatic potential examined in lungs by haematoxylin and eosin staining, and expression of ER, PR and HER2.

### Animal experiments

All *in vivo* experiments were performed in accordance with the Swiss animal welfare ordinance and approved by the cantonal veterinary office Basel-Stadt. Female NOD/SCID, NSG, and BALB/c mice were maintained in the Friedrich Miescher Institute for Biomedical Research and the Department of Biomedicine animal facilities in accordance with Swiss guidelines on animal experimentation. BALB/c, NOD/SCID and NSG mice originally are from the Janvier Labs and from in-house colonies. Mice were maintained in a sterile controlled environment (a gradual light–dark cycle with light from 7:00 to 17:00, 21–25°C, 45–65% humidity). For orthotopic engraftment of cancer cell lines in the limiting dilution assay, $250 \times 10^3$, $10 \times 10^3$, $2 \times 10^3$, 500, or 200 LM cells were suspended in 50 μl Matrigel:PBS (1:1) and injected into the fourth mouse mammary gland of six- to eight-week-old female NOD/SCID mice.

The frequencies of tumour-initiating cells in the different conditions were calculated and compared statistically using the Extreme Limiting Dilution Analysis (ELDA) online tool (Hu & Smyth, 2009). For the PB *in vivo* screen, HR1. Ctrl and HR1. PB cells ($1 \times 10^6$) were resuspended in 50 μl Matrigel:PBS (1:1) and injected into the mammary fat pads of four- to eight-week-old female NOD/SCID mice. For orthotopic engraftment of cancer cell lines, LB-mHR1 or LM cells ($250 \times 10^3$ cells) were resuspended in 50 μl Matrigel:PBS (1:1) and injected into the mammary fat pads of four- to eight-week-old female NOD/SCID mice. SUM159PT or 4T1 ($250 \times 10^3$ cells) cells were injected into the mammary fat pads of four- to eight-week-old female NSG or BALB/c mice, respectively. Tumours were measured with Vernier callipers and volume calculated as $0.5 \times ([\text{large diameter}] \times [\text{smallest diameter}]^2)$. Tumours were resected before they reached 1,500 mm³ and mice were monitored regularly for signs of metastatic outgrowth and distress. For survival studies, animals were sacrificed when tumours reached 1,500 mm³ or when they showed any signs of distress (e.g. breathing disorders, weight loss or immobility). All orthotropic experimental procedures (tumour resection and tumour cell implantation) were undertaken on anaesthetized mice by a single investigator according to protocols approved by the cantonal veterinary office Basel-Stadt. Experimental metastasis assays were performed by injecting $100 \times 10^3$ cells suspended in 100 μl of PBS into tail veins (*i.v.*). After intravenous injection of LM, LB-mHR1 or SUM159PT cells, *in vivo* bioluminescence imaging was performed to confirm injection and to monitor metastatic outgrowth. For bioluminescent imaging, mice were injected *i.p.* with 100 μl of D-luciferin (15 mg/ml, Biosynth L8220). Mice were anesthetized with isoflurane (2% in 1 l/min oxygen) and bioluminescence imaging performed using an IVIS Lumina XR instrument (Caliper LifeSciences) upon injection of luciferin. Acquisition times ranged from 3 to 10 min.

### Splinkerette PCR for the amplification of transposon integration sites

For PB sequencing, we adapted the splinkerette PCR protocol described previously (Friedrich *et al*, 2017). Tumour samples (16 primary tumours) and 18 metastatic samples (3 lung metastases, 6 lung macro-metastases and 9 LM cell lines) were sequenced. Genomic DNA was isolated using a DNeasy Blood and Tissue Kit (Qiagen) and sheared with a Covaris sonicator to a fragment length of 250 bp. After end repair and A-tailing, purified DNA fragments were ligated to a splinkerette adaptor (obtained after annealing of top 5'-gttcccatggtactactcatataatacgactcactataggtgacagcgagcgct-3' and bottom 5'-gcgctcgctgtcacctatagtgagtcgtattataattttttttttcaaaaaaa-3'). Transposon-containing fragments were enriched by 18 cycles of transposon-specific PCR1 for the 5' transposon ends in a unique library (5'-gacggattcgcgctatttagaaagagag-3' for the 5' arm of PB, and common splinkerette primer 5'-gttcccatggtactactcata-3'). Bar coding of individual samples and completion of Illumina adaptor sequences were achieved by an additional 12 cycles (primary tumours) and 18 cycles (metastatic samples) of transposon-specific PCR and a custom array of 35 unique bar-coding primers. For the 5' arm, we used 5'-aatgatacggcgaccaccgagatctacacatgcgtcaattttacgcagactatc-3' and for the splinkerette side, we used 5'-caagcagaagacggcatacgagatcggtXXXXXXXXtaatacgactcactatagg-3' primers. The Xs represent the bar code of eight nucleotides. After magnetic bead purification

(Beckman Ampure XP), libraries were assembled in two pools and sequenced on an HiSeq Desktop Sequencer (Illumina): HiSeq 2500 and MiSeq, Rapid Run, Paired-End, 2 × 100 bp, with 15% PhiX. A 7-pM aliquot was loaded on to the instrument.

## Mapping of insertion sequences to the mouse genome and identification of integration sites

### Pre-processing and alignment

Paired-End reads were first pre-processed by removing the expected transposon-derived sequence (5'-TAGGGTTAA-3') from the beginning of the first read (read pairs with non-matching first reads were discarded; typically around 1%) using the preprocessReads function from QuasR (version 1.12.0). Read pairs were then aligned to the mouse genome (BSgenome. Mmusculus. UCSC.mm10) using the QuasR qAlign function and parameters "-m 1 --best --strata --maxins 1000"; only uniquely mapping pairs with up to 1,000 bp between-pair distance were reported. Mapping rates were recorded and non-mapped read pairs were further aligned against the non-mobilized transposon sequence to estimate the probable fraction of read pairs that originate from non-mobilized copies of the transposon. For each aligned read pair, the *piggyBac* insertion coordinate was identified as the coordinate of the first (most 5') base of the first read.

### Integration site identification and quantification

For each unique integration site, the number of distinct supporting alignments (distinct read pairs), and the genomic sequence from the four base pairs on the same strand as the first read directly upstream of the insertion site were recorded. Only integration sites with the expected upstream TTAA sequence were used for the downstream analysis.

### Association of integration sites with genes

Coordinates of known genes (exons/introns, 5'-untranslated region [UTR], coding sequence [CDS], 3'UTR) were obtained from the TxDb. Mmusculus. UCSC.mm10.knownGene—Bioconductor package (version 3.2.2). Promoter regions were defined as regions 2,000 bp upstream of known transcript start sites. Integration sites were matched against these genomic regions to identify overlaps on any strand and orientation, selecting the first overlap in the case of multiple overlaps. Integration sites were classified hierarchically as follows: sites without overlaps to any transcript were labelled as promoter sites (in the case of an overlap with a promoter region) or intergenic sites. All other sites were labelled with the first region type that they overlapped, in the following order: 5'UTR, CDS, 3'UTR, intron or ncRNA (defined as an overlap with a transcript without annotated CDS). Sites were further labelled according to their orientation with respect to the associated gene (same or opposite). Finally, sites were grouped according to the gene they overlapped (including promoter sites) or, for intergenic sites, according to the pair of flanking genes.

### Selection of the enriched unique integration sites

The diversity has been calculated for each insertion, i.e., the number of distinct alignment pairs starting at the insertion coordinate (Chapeau *et al*, 2017). For each gene, the diversity was summed to give the number of unique insertions in a gene. Finally, this value was divided by the total number of unique insertions for a given sample. In order to find putative metastatic genes, we looked at genes with more unique insertions in the metastatic than in the tumour samples.

## Chromatin immunoprecipitation assay (ChIP)

Cells growing as monolayer cultures were fixed in 1% formaldehyde (SIGMA) for 10 min, quenched by addition of 0.125 M glycine (SIGMA), washed twice with 10 ml of PBS and 3.5 ml of SDS buffer (NaCl 100 mM, Tris–Cl pH8 50 mM, 5 mM EDTA pH 8.0, NaN$_3$ 0,02%, SDS 0,5%) including inhibitors (Complete Mini, ROCHE) were added. Lysed cells were incubated in IP buffer (100 mM Tris at pH 8.6, NaCl 100 mM, 0.5% SDS, 5% Triton X-100, and 5 mM EDTA with proteinase inhibitor) and the chromatin disrupted by sonication using a Diagenode Bioruptor sonicator UCD-300 to obtain fragments of 200–500 bp (15 cycles with 30 s of ON and 30 s OFF). For each ChIP, suitable amount of fragmented and pre-cleared chromatin was diluted and incubated with specific antibodies overnight. Antibodies used were normal rabbit IgG (Diagenode, C15410206) as a control and anti-NFIB (Sigma, HPA003956-100UL; 2 µl of 1 µg/µl dilution). Immunoprecipitated complexes were recovered on Dynabead Protein A beads (Thermo Fisher) and, after extensive washes, DNA was recovered by reverse crosslinking and purification using QIAquick PCR purification kit (Qiagen). For ChIP-quantitative PCR (qPCR), real-time PCRs were performed using Fast SYBR green master mix reagent (Applied Biosystems). Primer sequences used for ChIP-qPCR experiments were designed using Cistrome Data Browser (http://cistrome.org/db/), Integrated Viewer Genome software (Robinson *et al*, 2011) and data from GSE74826 (https://www.ncbi.nlm.nih.gov/geo/query/acc.cgi?acc = GSE74826; (Shin *et al*, 2016): Ero1l Peak 18492 FW 5'-GAGACTGCAGAGGGACAAGA-3', REV 5'-GCGCTCAGTTGAAACTCTGT-3'; Ero1l Peak 18493 FW 5'-AAAGACGCGGTCCTTCC-3', REV 5'-AAGGCTTAGGCAGCCAGA-3'; Meg3_TSS FW 5'-AAACAACGCTCTCCTTTCCTAAG-3', REV 5'-AAATAACCCCAACTGGTGATTG-3'.

## Genome editing by CRISPR-Cas9

A single sgRNA that produced a frameshift mutation in first and second exons was designed using the Zhang's lab online CRISPR (http://crispr.mit.edu) and cloned into a modified PX330 (Addgene plasmid 42230) in which the puromycin cassette was replaced by red fluorescent protein (RFP; provided by the group of M. Bühler at the Friedrich Miescher Institute, Basel). The sgRNA sequences selected for *Nfib* (based on the lowest number of predicted off-targets and highest predicted efficiency) were, guide 1: 5'-CACCG CTCCGGGAAAGTGCGTTTTA-3' and 5'-AAACTAAAACGCACTTTC CCGGAGC-3'; guide 2: 5'-CACCGTAGGCAATTGCACGGACG TG-3' and 5'-AAACCACGTCCGTGCAATTGCCTAC-3'. The sgRNA sequences selected for *Foxp1* were, guide1: 5'-CACCGCTTCGTGACA CTCGGTCCAA-3' and 5'-AAACTTGGACCGAGTGTCACGAAGC-3'; guide 2: 5'-CACCGTAGTAAGTGGTTGCCACCGC-3' and 5'-AAA CGCGGTGGCAACCACTTACTAC-3'. The sgRNA vectors were transduced into LM and 4T1 cells and the cells sorted for RFP positivity into 96-well plates. Single cell clones were expanded and screened by immunoblotting. Different numbers of clones were pooled in equal proportions to minimize undesired off-target and clonal effects (LM1 KO = 5, LM9 KO = 2, 4T1 WT = 2, 4T1 KO = 5). For human

*NFIB* overexpression, we used SAM-engineered Cas9 activation complexes (Konermann *et al*, 2015). LB-mHR1 and SUM159PT cells were lentivirus infected in order to express the full murine and human *NFIB* from an endogenous promoter and maintained in culture for two months. sgRNA sequences were designed using Zhang's lab online CRISPR (http://sam.genome-engineering.org) and cloned into the lentiviral vector Addgene; plasmid 61427. Lenti-sgRNA-MS2 was digested with BsmbI and purified in a gel. The two annealed primers were ligated using the golden gate reaction. The sgRNA sequences selected for mouse *Nfib* were, guide 2: 5'-CACCG CAGGAGGAGGAGGAGTAAAG-3' and 5'-AAACCTTTACTCCTCCT CCTCCTGC-3'; guide 3: 5'-CACCGTGGGGGAGGCGCGCGGGAGG-3' and 5'-AAACCCTCCCGCGCGCCTCCCCCAC-3'; guide 4: 5'-CACC GTGTGGAGAGGCTGGTGCAAA-3' and 5'-AAACTTTGCACCAGCC TCTCCACAC-3'. The sgRNA sequences selected for human *NFIB* were, guide 1: 5'-CACCGAGCTGAGCCATCCATTCCTC-3' and 5'-AA ACGAGGAATGGATGGCTCAGCTC-3'; guide 2: 5'-CACCGACTAGGC TTGCAGTAAACGC-3' and 5'-AAACGCGTTTACTGCAAGCCTAGTC-3'; guide 3: 5'-CACCGGAAGAGACTTGTCAGTATA-3' and 5'-AAACTA TACTGACAAGTCTCTCCC-3'.

## Lentiviral infections

The ATP1 vector was described previously (Rad *et al*, 2010). The transposon has PB inverted terminal repeats and can, therefore, be mobilized with the transposon system. ITRs were cloned into pBlue-Script and the following genetic elements introduced between the ITRs of the transposon: Carp β-actin splice acceptor (CβASA), En2SA splice acceptor from exon 2 of the mouse Engrailed-2 gene, Lun-SD from exon 1 of the mouse Foxf2 gene, and two bidirectional SV40 polyAs. Promoter elements carried by the transposons were unique to individual transposons: CAG (CMV immediate early enhancer and chicken beta-actin gene promoter) for ATP1. The *piggyBac* transposase was cloned out of the Super PiggyBac Transposase expression (System Biosciences) by PCR (primers 5' AGCTAGCACCGGTCGGAATTGTACCCAATTCGTTAAG 3' and 5' AGAATTCTTAATTAATTCTGGCGGCCGTTACG 3') and cloned into a doxycycline-inducible vector pFS250 (a kind gift from Novartis). For human *ERO1A* knockdown, we used two shRNA constructs (V2THS_85712 and V2THS_85710: Dharmacon pTRIPZ). Non-targeting shRNAs (pTRIPZ) were used as control. For mouse *Ero1l* knockdown, we used two shRNA constructs (MSH100691-LVRU6MH-a and MSH100691-LVRU6MH-b: GeneCopoeia), non-targeting shRNA (CSHCTR001-1-LVRU6MH) was used as control. For mouse *Ero1l* overexpression we used a pLenti-C-Myc-DDK-P2A-Puro (Origene, MR207416L3). A dual green fluorescent protein-luciferase 2 reporter (eGFP-Luc2; Liu *et al*, 2010) vector was used for *in vivo* bioluminescence imaging experiments. Lentivirus batches were produced using PEI transfection on 293T cells as previously described (Britschgi *et al*, 2017). The titre of each lentivirus batch was determined in 4T1, LM, FMI-LB-mHR1 and SUM159PT cells. Cells were infected overnight in the presence of polybrene (8 µg/ml). p250. PBase selection was performed with 500 µg/ml Neomicin G418 (InvivoGen) and applied 48 h after infection. ATP1 selection was performed with 2.5 µg/ml puromycin (Sigma) and applied 48 h after transfection. The SAM-engineered Cas9 activation complex (Konermann *et al*, 2015) consists of three lentiviral vectors: Lenti MS2-P65-HSF1_Hygro (Addgene plasmid 61426), Lenti-sgRNA-MS2_Zeo

(Addgene; plasmid 61427), and Lenti dCAS-VP64_Blast (Addgene 61425). Cells were infected first with a 1:1 mass ratio of Lenti MS2-P65-HSF1 and Lenti dCAS-VP64 at a MOI of 10 viral particles per cell; selection was carried out in 1 µg/ml Blasticidin (Gibco) and 500 µg/ml Hygromicin (InvivoGen) after a 24-h transduction. Selection in 500 µg/ml Zeomicin (Invitrogen) was also applied after a 24-h infection with Lenti-sgRNA-MS2_Zeo. Cells were cultured for two months *in vitro* before performing experiments.

## Transient gene silencing

The siRNA IDs were as follows: si*Nfib* (SI00178451, SI02668085 and SI02733983), (s4823 and s4825), non-targeting controls (D-001810-10-05) were ordered as onTarget-plus SMART pools (Dharmacon). Transfections of siRNAs were performed according to the manufacturer's guidelines (DharmaFect 1, Dharmacon).

## Fluorescence-activated cell sorting

For FACS, cell lines were detached using trypsin-EDTA, resuspended in growth medium and counted. Cells were passed through a 40-µm strainer (Falcon) and resuspended in PBS with 1% FCS. DAPI (0.2%, Invitrogen) was added (1:250) 2 min before cell sorting. Single cells were gated on the basis of their forward and side-scatter profiles and pulse-width was used to exclude doublets. Dead cells (DAPI bright) were gated out and RFP and GFP bright cells were selected. FACS was carried out with a BD FACSAria III (Becton Dickinson) using a 70-µm nozzle for SUM159PT, 4T1, HR1 and LM cell lines.

## Immunoblotting

Cells were lysed in RIPA buffer (50 mM Tris–HCl pH 8, 150 mM NaCl, 1% NP-40, 0.5% sodium deoxycholate, 0.1% SDS) supplemented with 1x protease inhibitor cocktail (Complete Mini, Roche), 0.2 mM sodium orthovanadate, 20 mM sodium fluoride, and 1 mM phenylmethylsulfonyl fluoride. Extracted proteins were measured using a DC protein assay kit (Bio-Rad #5000112) and their concentrations adjusted. Whole-cell lysates (30–50 µg) were subjected to 8% SDS–PAGE, transferred to PVDF membranes (Immobilon-P, Millipore) and blocked for 1 h at room temperature with 5% milk in PBS–0.1% Tween 20. Membranes were then incubated overnight with antibodies as indicated and exposed to secondary HRP-coupled anti-mouse or -rabbit antibody at 1:5,000–10,000 for 1 h at room temperature. For each of the blots presented, the results represent at least three independent experiments. The following antibodies were used: anti-NFIB (Sigma HPA003956-100UL, 1:1,000), anti-Vinculin (Abcam ab18058, 1:500), anti-ERK2 (Santa Cruz sc-1647, 1:2,000), anti-FOXP1 (Cell Signaling 4402, 1:1,000) and anti-ERO1L (Abnova H0003001-M01 Clone 4G3, 1:500).

## ELISA

Murine and human VEGFA protein levels were assessed using the mVEGFA Duo-set (R&D System, DY493-05) and hVEGFA Duo-set (R&D System, DY293B-05), respectively. Reagent Diluent Concentrate 2 (R&D Systems, DY995) was used. VEGFA ELISAs from R&D Systems were used according to the manufacturer's protocol.

## RNA isolation and qPCR

Total RNA was extracted using a Qiagen, RNeasy Plus Mini kit (cat. number 74136) according to the manufacture's protocol. Total RNA (1 µg) was transcribed using the iScript cDNA synthesis kit (Bio-Rad, cat. number 170-8891). PCR and fluorescence detection were performed using the StepOnePlus Sequence Detection System or the 7500 ABI Fast Cycler (Applied Biosystems) according to the manufactures' protocols in a reaction of 20 µl containing 1× TaqMan Universal PCR Master Mix (Applied Biosystems) and 25–50 ng cDNA. Primetime qPCT IDT assays (Integrated DNA technologies) were used for quantification of *Nfib* (Mm. PT.58.16634012), *Gapdh* (Mm. PT.39a.1), *Foxp1* (Mm. PT.58.2734907), *Ero1l* (Mm. PT.58.6905093), *Vegfa* (Mm. PT.58.1400306), *NFIB* (Hs. PT.58.40046929), *ERO1A* (Hs. PT.58.2554238), *VEGFA* (Hs. PT.58.1149801) and *HPRT1* (Hs. PT.58.45621572). All measurements were performed in technical duplicates and the arithmetic mean of the Ct values used for calculations: target gene mean $C_t$ values were normalized to the respective housekeeping genes (Hprt1 or Gapdh), mean $C_t$ values (internal reference gene, $C_t$), and then to the experimental control. The values obtained were $2^{-\Delta\Delta Ct}$ expressed as fold changes in regulation compared with the experimental control using the $2^{-\Delta\Delta Ct}$ method of relative quantification.

## Tumoursphere cultures

LM and HR1 cells were plated in ultra-low attachment plates (Corning) for six days at 10,000 cells per mL in DMEM:F12 supplemented with 1x B27 (Gibco, Invitrogen), 20 ng/ml human or mouse EGF (PeproTech), 20 ng/ml basic FGF (PeproTech) and 1× penicillin/streptomycin (Gibco, Invitrogen; Hilsenbeck *et al,* 2008). Primary tumourspheres were dissociated with 0.05% trypsin and replated at the same density for six further days for secondary tumoursphere assessment; additional rounds of culture were performed. All measurements were performed in technical triplicates.

## Matrigel-based tube formation assay

50 µl of growth factor reduced matrigel phenol red-free (Corning) were spread evenly in a 96-well plate, which was put in the incubator (37°C) for 30 min to allow matrigel to solidify. Then HUVEC cells (10–15,000 cells per well) were seeded on the matrigel in a total volume of 200 µl of conditioned medium from LM, 4T1, HR1 and SUM159PT cells. Recombinant VEGF₁₂₁ (PrepoTech 100-20A, 0.4 ng/µl) was added to the conditioned medium. After 4 h, cells were imaged using a ZEISS Axio Vert. A1 Inverted Microscope and finally fixed with 4% PFA. HUVEC network and number of master segments were quantified using the Angiogenesis Analyzer tool from ImageJ (Fiji) (Carpentier *et al,* 2020).

## Transwell migration and invasion assay

Migration assay was performed using transwell chambers 8 µm pore size (Corning) accordingly to the manufacturer's protocol. Cells were starved for 16 h, and 15,000 cells were seeded on the insert in 500 µl medium containing 0.5% FBS, and 750 µl of full growth medium including 10% FBS was added to the well. Cells were allowed to migrate towards the FBS gradient for 24 h before the

inserts were washed with PBS and remaining cells on the upper surface of the membrane were removed using cotton swabs. The cells that had migrated though the membrane were fixed with 4% PFA and stained with 0.2% Crystal violet. The whole membranes were then imaged using a ZEISS Axio Vert. A1 Inverted Microscope.

For the invasion assay, 1,000 cells were seeded in a 96-well plate on top of solidified growth factor reduced matrigel (Corning) in 200 µl assay medium (DMEM medium supplemented with 2% matrigel and 2% FBS). After 2 h at 37°C, 300 µl of DMEM medium with 10% matrigel and 2% FBS were added and the medium was replaced with 400 µl of assay medium every three days until the end of the experiment (10/12 days total). After two weeks, the number of invasive structures was counted.

## Sulphorhodamine-B (SRB) cell proliferation assay

1,000 cells per well were seeded in five 96-well plates and fixed after two-three hours or every consecutive day until day five in 100 µl of cold 3.3% trichloroacetic acid and incubated at 4°C for 1 h or O/N. The plates were washed four times with slow-running tap water and air-dried at RT. 100 µl of 0.057% sulphorhodamine B (SRB) dye were added to each well and plates were left at RT for 30 min, then rinsed four times with 1% (vol/vol) acetic acid and air-dried at RT. For optical density (OD) measurement, 200 µl of 10 mM Tris base solution (pH 10.5) were added and the OD was read using the Synergy H1 microplate reader (at 510 nm; BioTek).

## Circulating tumour cell quantification

Circulating tumour cells (CTCs) were isolated from the peripheral blood of animals bearing tumours just before tumour resection. Peripheral blood was plated in DMEM medium supplemented with 10% FCS and colonies counted on day 10 of culture. The number of CTCs was expressed as the total number of colonies in the dish divided by the volume of blood taken.

## RNA sequencing and analysis

### Tumour RNA preparation and sequencing

Sorted GFP-positive cells from tumours were collected and total RNA was extracted using a Qiagen, RNeasy Plus Mini kit (cat. number 74136) according to the manufacture's protocol. RNA integrity was measured on an Agilent 2100 Bioanalyzer using RNA Pico reagents (Agilent Technologies). The library was prepared using an Illumina TruSeq stranded mRNA-seq preparation kit (according to recommendations from the manufacturer). Library quality was measured on an Agilent 2100 Bioanalyzer for product size and concentration. Single-end libraries were sequenced with an Illumina HiSeq 2500 (50-nt read length).

### Tumoursphere RNA preparation and sequencing

Cells were dissociated with 0.05% trypsin, and total RNA extracted using a Qiagen, RNeasy Plus Mini kit (cat. number 74136) according to the manufacture's protocol. RNA integrity was measured on an Agilent 2100 Bioanalyzer using RNA Pico reagents (Agilent Technologies). The library was prepared using an Illumina TruSeq stranded mRNA-seq preparation kit (according to recommendations from the manufacturer). Library quality was

measured on an Agilent 2100 Bioanalyzer for product size and concentration. Single-end libraries were sequenced by a NextSeq 500 (76-nt read length).

### Analysis

Reads were aligned to the mouse genome (UCSC version mm10) with STAR (Dobin *et al*, 2013). The output was sorted and indexed with samtools. Stand-specific coverage tracks per sample were generated by tiling the genome in 20-bp windows and counting 5' end of reads per window using the function bamCount from the Bioconductor package bamsignals (https://bioconductor.org/packages/release/bioc/html/bamsignals.html). These window counts were exported in bigWig format using the Bioconductor package rtracklayer (https://bioconductor.org/packages/release/bioc/html/rtracklayer.html). The qCount function of QuasR (https://bioconductor.org/packages/release/bioc/html/QuasR.html) was used to count the number of reads (5' ends) overlapping the exons of each gene, assuming an exon union model. Differential gene expression analysis was performed using the limma-voom framework (Carriço *et al*, 2011).

### Cellular ROS quantification

To quantify ROS levels, we used the cellular reactive oxygen species detection assay kit (Deep Red fluorescence; Abcam, #ab186029). 20,000 cells were plated in a 96-well black wall/clear bottomed microplate and incubated ON. The day after, ROS Deep Read working solution was added for 30–60 min in the dark at RT. Wells with no dye solution were used as control. The deep red fluorescence was read using the Synergy H1 microplate reader (at excitation 650 nm, emission 675 nm; BioTek).

### Immunohistochemistry

All xenograft tissues were fixed in formalin fix (Shandon, Thermo Fisher) for 24 h at 4°C. Samples were then dehydrated, embedded in paraffin, sectioned (3–4 μm) and processed for haematoxylin/eosin staining and for immunohistochemistry. All immunohistochemistry experiments were performed using a Ventana DiscoveryXT instrument (Roche Diagnostics) following the Research IHC DAB Map XT procedure. For NFIB staining, slides were pretreated with CC1 for 90 min. NFIB primary antibody (1:250, Sigma-Aldrich, HPA003956) was incubated for 44 min, followed by secondary antibody incubation and detection. The RUO Discovery Universal procedure was used for CD31, VEGFA and ERO1A staining, with a CC1 pre-treatment (40 min) and incubation with a rat anti-CD31 (SZ31 dianova, 1:50), or with rabbit anti-VEGFA (GeneTex GTX61510, 1:100), or with rabbit anti-ERO1A (GeneTex, GTX112589 1:1,000) antibodies for 1 h at 37°C, respectively. Next, a polymer Immpress goat anti-rat for CD31 (Vector Lab, MP-7444) or Universal Immuno-enzyme Polymer (UIP) anti-rabbit (Nichirei, 414142F) for VEGFA and ERO1A was applied for 28 min at 37°C, and the peroxidase reaction revealed with the Discovery ChromoMap DAB kit (Ventana, Roche diagnostics). Counter staining was performed with haematoxylin II and bluing reagent (Ventana, Roche diagnostics). For PDXs, the overall procedure was the same but the signal was amplified using the amplification kit (Ventana, Roche diagnostics, 760-080). Two

images of two to eight lung metastases, two images for tumours and PDX of each condition were captured and quantified manually or with ImageJ (Fiji).

### CD31, NFIB, ERO1A and VEGFA quantification

The images were analysed using Fiji open source software (Schindelin *et al*, 2009). First, the lung area for each tissue section was measured by manual selection. Second, for each metastasis in each tissue section, a region of interest (ROI) was selected manually. Each ROI was then subjected to background subtraction (rolling ball with a 50-px radius) followed by colour deconvolution (Ruifrok & Johnston, 2001) using [H DAB] vectors. The resulting colour channel (2) corresponding to the DAB stain was blurred (Gaussian Blur with a sigma of 5) and was then thresholded (using the value for all images [222]) to measure the CD31-positive areas. For NFIB, ERO1A and VEGFA IHC images, stained tissues areas were selected manually and the pixels quantified by colour deconvolution ([H DAB] vectors) and thresholding the colour channel (2) (using a value of [170] for all images). Scripts for these semi-automatic analyses are available upon request.

### Immunofluorescence

12 mm glass slides were coated with 200 μl of 0.1 mg/ml Poly-L-Lysine (SIGMA) for two hours at 37°C, washed twice with PBS, and 10,000 cells seeded per well. The day after, slides were washed once with ice-cold PBS and fixed with 4% PFA 4% for 15 min and washed three times 5 min with ice-cold PBS, then permeabilized with Triton 0.3% PBS for 15 min at RT, and washed three times 5 min with ice-cold PBS. Then cells were blocked with 2.5 % normal goat serum (NGS) in PBS for one hour at RT, incubated with the primary antibody in 2.5 % NGS in PBS ON at 4°C or for two hours at RT on rocking table. Next, cells were washed three times 5 min with ice-cold PBS and incubated with the secondary antibody 2.5 % NGS in PBS for 2 h at RT on rocking table in the dark. After three times 5-min washing in PBS, the nuclei were stained with DAPI (2 mg/ml; Invitrogen) for 10 min at RT, after two more washes for 5 min with PBS, the slides were dried and mounted with one drop Prolong Gold anti-fade mounting medium (Invitrogen), and 24 h later the slides were analysed or stored at 4°C. The following antibodies were used: anti-rabbit HIF1a (Abcam ab2185, 1:1,000) and anti-rabbit Alexa 633 (Thermo Fisher, 1:1,000).

### Microscopy image acquisition

For immunohistochemistry sections, images were captured using a Nikon Ni-e upright microscope coupled to a PRIOR slide loader. Acquisition was performed with a 4× AIR objective with a DS-Fi3 camera using NIS software. Quantification of the images was performed using Fiji. Phase-contrast imaging used an inverted Zeiss Axioscope (10×, NA = 0.25 and 5×, NA = 0.15) equipped with an Axiocam 503 mono 60N-C camera (pixel size 4.54 μm) and images were acquired using the Zen lite software. For immunofluorescence, images of stained sections were captured using Nikon T2 microscope, with a Photometrics Prime 95B camera using NIS software.

## Data analysis from publicly available datasets

Mutation, segmented DNA copy number and mRNA expression data of breast tumours were from METABRIC (Curtis *et al*, 2012; Pereira *et al*, 2016) and TCGA Research Network (Koboldt *et al*, 2012) from cBioPortal (Cerami *et al*, 2012; Gao *et al*, 2013). We renamed the segmented DNA copy number data levels (produced using GISTIC 2.0) from −2, −1, 0, 1, 2 to "Deep Deletion", "Deletion", "Diploid", "Gain", and "High-level Amplification", respectively. Relapse-free survival, distant metastasis-free survival and overall survival of *NFIB* and *ERO1A* were generated using the 2017 version of KMplotter (Győrffy *et al* 2010) (http://kmplot.com/analysis/index.php?p = service&cancer = breast). Venn diagrams were produced using Bioinformatics & Evolutionary Genomics http://bioinformatics.psb.ugent.be/webtools/Venn/.

## Statistical data analysis

The standard laboratory practice randomization procedure was used for cell-line groups and animals of the same age and sex. The investigators were not blinded to allocation during experiments and outcome assessment. The number of mice was calculated by power analysis using data from small pilot experiments. Values represent the means ± s.d. unless stated otherwise. $P$ values were determined using unpaired two-tailed $t$-tests and statistical significance set at $P = 0.05$. The variance was similar between the groups compared. Biological replicates correspond to different cell lines and tumour material. Technical replicates are tests or assays run on the same sample multiple times. The means of technical replicates, if available, were used for analysis and visualization. Biological replicates are tests or assays run on different samples and were used for statistical analysis and for reporting the number of experimental entities. Data were tested for normal distribution, Student's $t$-tests and two-way ANOVA (if normally distributed) or nonparametric Mann–Whitney $U$-test or Wilcoxon tests were applied unless stated otherwise. Kaplan–Meier plots were generated using the survival calculation tool from GraphPad Prism and significance was calculated using the two-tailed log-rank test at $P < 0.05$. For the analysis of tumour growth, we extracted for each individual sample the time to reach a tumour volume of 250 mm$^3$ as follows: a linear model was fit to the log of the tumour volume as a function of time using all volumes greater than zero and least squares fitting in R (lm function). The resulting fit typically had an $R$-squared of 0.7 or greater. The estimated times were then used to compare conditions, either in a two-way ANOVA or Student's $t$-test, as indicated.

# Data availability

- Tumour mRNA-seq data (GEO accession GSE144392): https://www.ncbi.nlm.nih.gov/geo/query/acc.cgi?acc = GSE144392.
- Tumoursphere mRNA-seq data (GEO accession GSE144393): https://www.ncbi.nlm.nih.gov/geo/query/acc.cgi?acc = GSE144393.
- PiggyBac screen genomic GEO accession (GSE144898): Go to https://www.ncbi.nlm.nih.gov/geo/query/acc.cgi?acc = GSE144898.

**Expanded View** for this article is available online.

### The paper explained

**Problem**

Metastasis is the main cause of solid cancer-related death and is a major clinical challenge in breast cancer. Although many studies have thoroughly characterized and classified breast tumours, the molecular determinants of metastatic colonization remain elusive. Their delineation and functional validation are urgently needed to improve our understanding of this currently incurable disease.

**Results**

We engineered a non-metastatic cell line with the doxycycline (dox)-inducible *piggyBac* (PB) transposon mutagenesis system and performed an unbiased *in vivo* PB-mutagenesis genetic screen to identify drivers of metastatic colonization. We have shown that the transcription factor nuclear factor IB (NFIB) is necessary and alone sufficient for breast cancer metastasis. NFIB is upregulated in mammary tumourspheres and metastases. Transcriptional profiling of tumours and tumourspheres derived from highly metastatic cell lines and NFIB ChIP experiments showed that the lethal effect of NFIB is mediated by increased expression of the oxidoreductase *ERO1A*. Mechanistically and functionally, downregulation of *ERO1A* in *NFIB*-overexpressing models decreased ROS levels and HIF1α-VEGFA-mediated angiogenesis, reduced metastases and prolonged overall survival of the animals. Furthermore, *NFIB/ERO1A/VEGFA* co-expression correlates with the metastatic potential in triple-negative breast cancer (TNBC) patient-derived xenograft (PDX) models.

**Impact**

This study provides new molecular insights into the determinants of metastatic colonization. Our work not only highlights the power of genetic screening to identify functionally relevant metastatic networks, but also describes the mechanism by which the transcriptional factor NFIB mediates metastatic colonization. Thus, we have revealed a targetable network that influences colonization, the last and fatal step of the metastatic cascade.

## Acknowledgements

We thank members of the Bentires-Alj laboratory for advice and discussions. The authors are grateful for the support of the Genomics Core Facility at FMI and the BSSE-ETHZ, to the Animal facilities of the University of Basel and FMI. We thank S. Bichet (FMI) for help with immunohistochemistry, G. Galli for discussion and advice, S. Smallwood and S. Dessus-Babus for optimizing and running the PB-seq experiments. We thank A. L. Welm (University of Utah) for the PDX models. We thank C. Kuperwasser and S. Ethier for the SUM159 cells. This research and also the publication fees were funded by the Swiss National Science Foundation (#310030_184673 20). Research in the Bentires-Alj laboratory is supported by the Swiss Initiative for Systems Biology—SystemsX, the European Research Council (ERC advanced grant 694033 STEM-BCPC), Novartis, the Krebsliga Beider Basel, the Swiss Cancer League, the Swiss Personalized Health Network (Swiss Personalized Oncology driver project), and the Department of Surgery of the University Hospital Basel.

## Author contributions

Study conception and manuscript writing: FZ, PMR, MB-A; Design and experiments, data analysis and result interpretation: FZ, PMR Computational analysis of RNA-seq, data analysis and result interpretation: P-AM; Intersection between RNA-seq data and Nfib putative targets, data analysis and result interpretation: CJ; Computational analysis of RNA-seq and PB-seq experiments, data analysis and result interpretation: AS, MBS; Performing and optimizing immunohistochemical staining, data analysis and result interpretation: M-MC; PB library preparation and quality control for PB-seq: TE; Algorithm for

immunohistochemistry and quantification: LS; LB-mHR1 cell line: LB; Experiments for the generation of HR1-PB cell lines: JPC; PBase and ATP1 vectors and conceptual development of the project: RR; Conceptual development of the project: MRJ; Endothelial cells and reagents and conceptual development of the project: AR and AB; Reading and approval of the final manuscript: All authors.

## Conflict of interest

P. M. R., L. B., J. P. C. and M. R. J. are employees of Novartis Pharma AG, M. B.-A. owns equities in, and receives laboratory support and compensation from Novartis, and serves as consultant for Basilea. The other authors declare that they have no conflict of interest.

## For more information

i    TCGA: https://portal.gdc.cancer.gov/.

ii   GSEA: http://software.broadinstitute.org/gsea/index.jsp.

iii  Kaplan–Meier plotter: http://kmplot.com/analysis/.

iv   cBioportal: https://www.cbioportal.org/.

v    Bioinf. & Evol. Gen: http://bioinformatics.psb.ugent.be/webtools/Venn/.

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
