## [Review Process File · EMBO Molecular Medicine]

The NFIB-ERO1A axis promotes breast cancer metastatic colonization of disseminated tumour cells

Federica Zilli, Pedro Ramos, Priska Auf der Maur, Charly Jehanno, Atul Sethi, Marie-May Coissieux, Tobias Eichlisberger, Loic Sauter, Adelin Rouchon, Laura Bonapace, Joana Couto, Roland Rad, Michael Rugaard Jensen, Andrea Banfi, Michael Stadler, and Mohamed Bentires-Alj

DOI: [10.15252/emmm.202013162](https://doi.org/10.15252/emmm.202013162)

Corresponding author: Mohamed Bentires-Alj (m.bentires-alj@unibas.ch)

Review Timeline:

Submission Date:	22nd Jul 20
Editorial Decision:	1st Sep 20
Revision Received:	16th Dec 20
Editorial Decision:	12th Jan 21
Revision Received:	29th Jan 21
Accepted:	1st Feb 21

Editor: Lise Roth

Transaction Report:

1st Sep 2020

Thank you for the submission of your manuscript to EMBO Molecular Medicine, and please accept my apologies for the delay in getting back to you. We have now received feedback from the three reviewers who agreed to evaluate your manuscript. As you will see from the reports below, the referees acknowledge the interest of the study and are overall supporting publication of your work pending appropriate revisions. They nevertheless raise a number of concerns, particularly regarding the mechanistic aspect of the study that should be strengthened.

Addressing the reviewers' concerns in full will be necessary for further considering the manuscript in our journal, and acceptance of the manuscript will entail a second round of review. EMBO Molecular Medicine encourages a single round of revision only and therefore, acceptance or rejection of the manuscript will depend on the completeness of your responses included in the next, final version of the manuscript. For this reason, and to save you from any frustrations in the end, I would strongly advise against returning an incomplete revision.

When submitting your revised manuscript, please carefully review the instructions that follow below. Failure to include requested items will delay the evaluation of your revision:

- 1) A .docx formatted version of the manuscript text (including legends for main figures, EV figures and tables). Please make sure that the changes are highlighted to be clearly visible.
- 2) Individual production quality figure files as .eps, .tif, .jpg (one file per figure).
- 3) A .docx formatted letter INCLUDING the reviewers' reports and your detailed point-by-point responses to their comments. As part of the EMBO Press transparent editorial process, the point-by-point response is part of the Review Process File (RPF), which will be published alongside your paper.
- 4) A complete author checklist, which you can download from our author guidelines (<https://www.embopress.org/page/journal/17574684/authorguide#submissionofrevisions>). Please insert information in the checklist that is also reflected in the manuscript. The completed author checklist will also be part of the RPF.
- 5) Please note that all corresponding authors are required to supply an ORCID ID for their name upon submission of a revised manuscript.
- 6) Before submitting your revision, primary datasets produced in this study need to be deposited in an appropriate public database (see <https://www.embopress.org/page/journal/17574684/authorguide#dataavailability>). Please remember to provide a reviewer password if the datasets are not yet public. The accession numbers and database should be listed in a formal "Data Availability" section

(placed after Materials & Method). Please note that the Data Availability Section is restricted to new primary data that are part of this study.

7) We would also encourage you to include the source data for figure panels that show essential data. Numerical data should be provided as individual .xls or .csv files (including a tab describing the data). For blots or microscopy, uncropped images should be submitted (using a zip archive if multiple images need to be supplied for one panel). Additional information on source data and instruction on how to label the files are available at .

8) Our journal encourages inclusion of *data citations in the reference list* to directly cite datasets that were re-used and obtained from public databases. Data citations in the article text are distinct from normal bibliographical citations and should directly link to the database records from which the data can be accessed. In the main text, data citations are formatted as follows: "Data ref: Smith et al, 2001" or "Data ref: NCBI Sequence Read Archive PRJNA342805, 2017". In the Reference list, data citations must be labeled with "[DATASET]". A data reference must provide the database name, accession number/identifiers and a resolvable link to the landing page from which the data can be accessed at the end of the reference. Further instructions are available at .

9) We replaced Supplementary Information with Expanded View (EV) Figures and Tables that are collapsible/expandable online. A maximum of 5 EV Figures can be typeset. EV Figures should be cited as 'Figure EV1, Figure EV2' etc... in the text and their respective legends should be included in the main text after the legends of regular figures.

- Additional Tables/Datasets should be labeled and referred to as Table EV1, Dataset EV1, etc. Legends have to be provided in a separate tab in case of .xls files. Alternatively, the legend can be supplied as a separate text file (README) and zipped together with the Table/Dataset file. See detailed instructions here:

10) For more information: There is space at the end of each article to list relevant web links for further consultation by our readers. Could you identify some relevant ones and provide such information as well? Some examples are patient associations, relevant databases, OMIM/proteins/genes links, author's websites, etc...

11) Every published paper now includes a 'Synopsis' to further enhance discoverability. Synopses are displayed on the journal webpage and are freely accessible to all readers. They include a short stand first (maximum of 300 characters, including space) as well as 2-5 one-sentences bullet points that summarizes the paper. Please write the bullet points to summarize the key NEW findings. They should be designed to be complementary to the abstract - i.e. not repeat the same text. We encourage inclusion of key acronyms and quantitative information (maximum of 30 words / bullet point). Please use the passive voice. Please attach these in a separate file or send them by email, we will incorporate them accordingly.

Please also suggest a striking image or visual abstract to illustrate your article. If you do please provide a png file 550 px-wide x 400-px high.

12) As part of the EMBO Publications transparent editorial process initiative (see our Editorial at <http://embomolmed.embopress.org/content/2/9/329>), EMBO Molecular Medicine will publish online a Review Process File (RPF) to accompany accepted manuscripts.

In the event of acceptance, this file will be published in conjunction with your paper and will include the anonymous referee reports, your point-by-point response and all pertinent correspondence relating to the manuscript. Let us know whether you agree with the publication of the RPF and as here, if you want to remove or not any figures from it prior to publication.

I look forward to receiving your revised manuscript.

With my best wishes,

Lise

Lise Roth, PhD
Editor
EMBO Molecular Medicine

To submit your manuscript, please follow this link:

Link Not Available

Photos 400-800 DPI

Figures are not edited by the production team. All lettering should be the same size and style; figure panels should be indicated by capital letters (A, B, C etc). Gridlines are not allowed except for log plots. Figures should be numbered in the order of their appearance in the text with Arabic numerals.

Each Figure must have a separate legend and a caption is needed for each panel.

*Additional important information regarding figures and illustrations can be found at <http://bit.ly/EMBOPressFigurePreparationGuideline>

***** Reviewer's comments *****

Referee #1 (Remarks for Author):

In this manuscript, Zilli, Marques Ramos et al suggest that NF1B promotes breast cancer metastasis. In particular, the authors suggest that this transcription factor supports metastasis by means of ERO1A, which in turn modifies VEGFA in the tumor milieu. The authors used a previously developed mouse breast cancer cell line (PIK3CAH1047R driven) and an unbiased transposon mediated screening approach (piggyBac) to fish out genes that specifically support metastatic but not primary tumor colonization. Remarkably, this is an in vivo screening based on a phenotypic metastatic gain of function. The authors purify a series of PB driven clones with enriched metastatic capacity. Subsequently they sequence these clones and identify NF1B transcription factor as a mediator of metastatic potential. In a series of well-executed experiments, they functionally validate the role of NF1B by means of gain and loss of function in both mouse and human xenograft systems in complete or incomplete immune system. Next, they identify ERO1A as a potential downstream mediator, genetically validate it and link it to changes in VEGFA function. Whereas the earlier sections of the manuscript are very strong, the latter ones re ERO1A and VEGFA are less robust and would benefit from further elaboration to provide a clear mechanistic insight.

The function of NF1B in breast cancer metastasis as described is unclear. Although it is known as a Transcription Factor its function as such is not well defined in the current model system. In this sense, it is unclear how it operates to induce ERO1A. Is it direct binding at ERO1A promoter or enhancers? Is it indirectly by means of an activation of another intermediate? Alternatively, does it operate by other means.

Further, it is unclear how in basal like breast cancer NF1B levels are accumulated. Is the gene located in a commonly amplified region? Is it frequently mutated? Are there any signaling pathways that upon misregulation cause accumulation of NF1B.

Another mechanistic relevant point is how ERO1A operates to support metastasis. As described, it is genetically relevant but mechanistically it is unclear how an oxidoreductase located at the endoplasmic reticulum is functioning to support metastasis. Is ERO1A catalytic activity necessary? Finally, the relationship with VEGFA is referred in the literature but the current depicted data in the manuscript reads largely correlative and not causative.

Overall, there is the perception that further development on the mechanistic part is needed to robustly explain when and how NF1B is overexpressed in breast cancer, how it mediates ERO1A expression and how the latter supports colonization.

Several plots are based on small numbers (n=3 or n=4). I wonder how powered are these analyses and how many times repeated. Although the magnitude of the effect may seem large, this relies on

1 or 2 mice. Please comment.

The luminescent/fluorescent mice images shown are depicted using different intensity range. Please homogenize the range across figures as otherwise is confusing. Some small lesions seem huge when displayed, whereas some big lesions happen to be displayed as weaker. This is somewhat confusing and in addition may lead to misleading interpretations. What is the criteria to define a metastasis in each experimental setting, a signal above background?

Minor points

1- In figure 3C it is unclear what lines the p-value represents. Please address

2- Some of the KM plots in the last figure are not significant. Whereas a trend in the clinics may be used in some occasions here is pointless. Indeed, this would suggest that VEGFA is not related, clinically speaking, to ERO1A and NF1B as it negatively impacts on the magnitude of the genes association with the clinical outcome.

Referee #2 (Comments on Novelty/Model System for Author):

The initial model is interesting, however the choice of cell models to follow up in down-stream experiments (or more to the point, which ones are excluded) is curious and needs further consideration. The data needs more depth of support to warrant publication in a journal such as EMBO Mol Med.

Referee #2 (Remarks for Author):

The paper by Zilli et al., provides compelling evidence that NFIB promotes metastasis in breast cancer using a novel unbiased Piggy bac screen. These are important observations and is commensurate with an emerging role for NFIB as a driver of invasion and metastasis in other cancers including SCLC and melanoma. The support for NFIB as a metastatic driver is thorough and well supported, however the subsequent elucidation of the down-stream players in this metastatic/colonization axis are perhaps somewhat over-interpreted in the context of the data available. Particularly the pivotal role of ERO1A in facilitating the NFIB promotion of metastasis, particularly the data presented in figure 5 and the contention that NFIB-ERO1A induces angiogenesis. These conclusions need additional supporting data to be able to confidently draw some of the conclusions made with respect to the NFIB-EROA1-VEGFA axis. Over-all this is a good manuscript, with important and novel observations, that I think is worthy of being published in EMBO Molecular Medicine. However, there are some issues with respect to clarity and breadth and depth of data that need to be addressed, some of which I have commented on below.

General comments.

Interestingly NFIA also appears in the Piggy Bac screen however is not really commented on with the focus falling to NFIB (not surprisingly as it is statistically significant) and FOXP1 (which isn't statistically significant. Given that NFIA is a related family member it is probably worth noting it's appearance in the dataset. Is there a potential functional redundancy between family members?

Some clarification is needed with respect to the cell line pools used in these studies. Were the growth characteristics of each line analysed and found to be similar? ie when they are pooled are we truly

seeing the effect of the oligo pool or dominance by one or two of the sub-lines? In figure EV4 were the g3 and g4-NFIB lines pooled and injected into 16 mice, or individually into mice? I suspect the former, but were these lines analysed independently at all? What is the rationale in choosing cells to pool? For example EVFig 2 shows 5 lines of LN1 kd were used to create oligo pools but only 2 lines for the LM9. Was there a rationale behind excluding other LM9 lines?

In EVFig 3A clones 5 and 7 were used as a WT control however the larger NFIB isoform around 50kDa is convincingly ablated. Are these cells an appropriate control to use here? Were the parental cells used as a comparator as well?, as based on the western data in EVFig 3A they don't appear to be an appropriate control line.

Specific Comments

Figure 5. Identification of a novel NFIB down-stream target in ERO1A is novel and is certainly supported by the western blot analysis in EVFig 5C across the cell lines used in the study. It is curious though that mRNA analysis for validation is then restricted to the LM1/9 k.o cells. Why not include all lines analysed at the protein level? Are the mRNA changes consistent with the protein changes in all cell models?

Figure 5. Similarly, it is unclear why such a crucial observation as the significant role ERO1A plays in NFIB driven metastasis is only followed up in the SUM159PT cell line. Given that the data contends that the promotion of metastasis by NFIB is almost exclusively reliant on the ERO1A down-stream target it would be important to confirm this relationship in more than a single cell line model, even if the gene expression data supports the control of ERO1A expression across multiple lines. This observation needs to be followed up in additional lines used here.

Figure 5. I think the link with angiogenesis is interesting and can be suggested from some of the data but I feel the data lacks depth and strength to draw a strong conclusion that NFIB-ERO1A induces angiogenesis, as the section title in the results states and the discussion affirms. The VEGFA link is largely drawn from the previous reports by Tanaka, 2016, but the results state that "VEGFA expression increased in cells over expressing NFIB and decreased in cell lines and lung mets upon ERO1A down-regulation". However, again data is only shown here for the SUM159PT line, the analysis only confirms mRNA changes, there is no evidence that protein levels change, no VEGFA staining in the tumor sections. Was VEGFA expression altered in the RNA-seq analysis that identifies ERO1A downstream of NFIB? Was VEGFA expression altered in all of the cell line models reported here at both RNA and protein levels?

Figure 5. The interpretation of NFIB-ERO1A playing a role in angiogenesis needs more support than the IHC of lung metastases shown in Figure 5E. Were primary tumor samples analysed for CD31 staining? Additional angiogenesis data is needed and ideally using an alternative angiogenesis assay in addition to the inferences drawn from tumour sectioning shown here. Will conditioned media from NFIB over-expression (and related eg ERO1A k.d) models modulate endothelial tube formation? Is VEGFA truly the culprit here if this is the case? What about VEGFC?

Figure 6: Much of this data is drawn from bio-informatic analysis of RNA based data sets and provides some interesting correlations. Despite compelling supportive evidence from the cell line models, it is important to confirm this correlation at the protein level using human tumor samples. Is there a correlation between expression of NFIB, ERO1A and VEGFA in breast cancer tumor sections? Is there significant heterogeneity in breast cancer tumors whereby a subset with a propensity to metastasis are marked by elevated NFIB and down-stream targets highlighted in the

manuscript?

Referee #3 (Comments on Novelty/Model System for Author):

Adequate model systems were used in the manuscript including both experimental and spontaneous metastasis models to validate the role of NFID and ERO1A in breast cancer metastasis.

Referee #3 (Remarks for Author):

In this study by Zilli et al, the authors take a comprehensive in vivo approach (transposon insertional mutagenesis system) to screen a unique factor that promotes tumor metastasis and identifies a transcription factor nuclear factor IM (NFIB) is enhanced in highly metastatic murine mammary cancer clones. The importance of NFIB in tumor growth and metastasis was validated by gain- and loss-of-function studies in mice. The authors claim that NFIB contributes to metastasis by managing gene level of an oxidoreductase ERO1A, that manages VEGFA gene expression that contributes to angiogenesis and consequently promotes metastasis in mammary cancer. The authors also validate the clinical relevance of their finding in patients with triple negative breast cancer by analyzing publicly available database. Overall, this study has shown that NFIB, ERO1A, and VEGFA are critical factors for tumor progression in breast cancer, which is interesting and relevant to the cancer community. However, several issues should be addressed through additional experiments and clarification to provide firmer mechanistic evidence in support of the major conclusions.

1. Throughout the manuscript, the authors claim that the effect of NFID on tumor progression is mediated by increased expression of ERO1A. Authors also suggest that the NFID-ERO1A-VEGFA signaling axis is a metastasis driver. However, additional data are needed to strengthen this claim. ERO1A is chosen as a transcription target of NFID based on previous study done in different settings and the current data only shows that knockout of ERO1A decrease metastasis in vivo. This does not demonstrate that ERO1A is a significant downstream effector of NFID or a direct transcription target of NFID. The authors need to overexpress ERO1A in NFID ablated cells and see whether the phenotype lost by NFID knockout is rescued by ERO1A. In addition to data demonstrating that NFID knockout results in ERO1A expression decrease, the authors should provide additional evidence that NFID is a transcription factor of ERO1A in mammary cancer cells, for instance through CHIP and promoter activity assays in mHR1 or 4T1 cells.

2. On a related note, connection between NFID-ERO1A and VEGFA is not clear. The authors show that ERO1A knockout results in decreased gene expression of VEGFA. To clearly state that VEGFA is the last effector molecule in the NFID-ERO1A-VEGFA signaling axis, the authors should additionally knockout NFID and show the decreased gene and protein expression of VEGFA. It is also worth to know whether the overexpression of ERO1A in cells with NFID loss, not only results in enhanced metastasis potential but also restored VEGFA expression. In addition, the authors should at least discuss how ERO1A governs gene expression of VEGFA.

3. NFID has been implicated in tumor metastasis of breast cancer in previous studies and a proposed mechanism is through suppression of CDKN2A (PMID 30350349). Whereas, the authors suggest that NFID works through ERO1A in breast cancer progression. It is worth to know whether

which factor serves as a predominant downstream effector of NFID or whether they coordinately contribute to NFID mediated breast cancer metastasis.

4. In line with the comment above, the data in current manuscript are all originated from murine cell lines including LB-mHR1 and 4T1. The factors identified could be specific in murine models. The authors should validate the role of these key factors in human breast cancer cells.

5. The importance of NFID in tumor growth is demonstrated in Figure 2. The role of NFID in tumor metastasis is shown in Figures 3 and 4. The study focuses on validating NFID level and function using highly metastatic LM1,9, and 8 clones. It is unclear whether the main contribution of NFID is through promoting tumor proliferation or angiogenesis, or even through migration and invasion as others suggest in melanoma (PMID: 28119061). At least, in vitro studies validating and comparing the effect of NFID modulation on proliferation, angiogenesis, migration, and invasion using mHR1 cells are needed.

6. Figure 6 provides clinical relevance demonstrating that NFID or/and ERO1A and VEGFA levels correlate with worse clinical outcome. Although these data provide evidence that each factor are linked with patient survival, the connections between NFID, ERO1A, and VEGFA are not proven. Correlation study linking each factor to one another is needed. In addition, the analyzed data is limited to triple negative breast cancer patients. It is worth to know whether the NFID signaling is unique for breast cancer or it is a common critical tumor promoting factor for other solid tumor types.

7. Minor suggestions for expanded view (EV) figures are listed below.

EV 1C-1D: Tumor volume and latency of 9 groups are shown in each graph which are difficult to distinguish. The tumor volume and latency should be additionally shown as bar graphs as shown in EV Figure 1E for metastasis potential.

EV 4A: NFID expression in two NFID knock-in clones (g3 and g4) are shown. Lanes 2-3 are g3 clones and lanes 4-5 are g4 clones. However, the expression level of NFID differ between lanes (lane 2 vs 3, lane 4 vs 5). The authors need to clarify.

EV 5: This should be moved to main figures as it is a critical data showing the connection between NFID and ERO1A.

Manuscript # EMM-2020-13162- Point-by-point reply to the Reviewers

The NFIB-ERO1A axis promotes breast cancer metastatic colonization of disseminated tumour cells

Federica Zilli^{*}, Pedro Marques Ramos^{*}, Priska Auf der Maur, Charly Jehanno, Atul Sethi, Marie-May Coissieux, Tobias Eichlisberger, Loïc Sauter, Adelin Rouchon, Laura Bonapace, Joana Pinto Couto, Roland Rad, Michael Rugaard Jensen, Andrea Banfi, Michael B. Stadler and Mohamed Bentires-Alj

We genuinely thank the Reviewers for their comments and suggestions; they have helped us to improve the quality of our manuscript. The bold parts in the text refer to new figures and panels.

Point-by-point Reply to Reviewer #1:

Reviewer# 1: *In this manuscript, Zilli, Marques Ramos et al suggest that NFIB promotes breast cancer metastasis. In particular, the authors suggest that this transcription factor supports metastasis by means of ERO1A, which in turn modifies VEGFA in the tumor milieu. The authors used a previously developed mouse breast cancer cell line (PIK3CAH1047R driven) and an unbiased transposon mediated screening approach (piggyBac) to fish out genes that specifically support metastatic but not primary tumor colonization. Remarkably, this is an in vivo screening based on a phenotypic metastatic gain of function. The authors purify a series of PB driven clones with enriched metastatic capacity. Subsequently they sequence these clones and identify NFIB transcription factor as a mediator of metastatic potential. In a series of well-executed experiments, they functionally validate the role of NFIB by means of gain and loss of function in both mouse and human xenograft systems in complete or incomplete immune system. Next, they identify ERO1A as a potential downstream mediator, genetically validate it and link it to changes in VEGFA function. Whereas the earlier sections of the manuscript are very strong, the latter ones re ERO1A and VEGFA are less robust and would benefit from further elaboration to provide a clear mechanistic insight.*

Authors' response: We thank the Reviewer for this accurate summary of our findings. In this revised manuscript we performed experiments to strength the last section and provide a clear mechanistic insight.

Reviewer# 1: *The function of NFIB in breast cancer metastasis as described is unclear. Although is known as a Transcription Factor its function as such is not well defined in the current model system. 1) In this sense, it is unclear how it operates to induce ERO1A. Is it direct binding at ERO1A promoter or enhancers? is it indirectly by means of an activation of another intermediate? Alternatively, does it operate by other means.*

Authors' response: We are grateful to the Reviewer for these questions. To address how NFIB induces ERO1A expression, we performed chromatin immunoprecipitation (ChIP) for NFIB in our murine cell lines (LM1/LM1 *Nfib*KO, LM9/LM9 *Nfib*KO, 4T1 WT, 4T1 *Nfib* KO, HR1gCtrl and HR1g4*Nfib*) and found that ERO1L is a direct target of Nfib. Particularly, we extracted the sequence of ERO1L peaks from published ChIP-seq data of *Nfib* in the mammary gland (Shin et al., 2016, acc: GSE42900) using Cistrome Data Browser (<http://cistrome.org/db>), and Integrated Viewer Genome software (J. T. Robinson et al. 2011) and designed primers on the ERO1L promoter region peaks. Notably, qPCR analysis revealed increased NFIB recruitment in all the *Nfib* overexpressing cells (LM1, LM9, 4T1 WT and HR1g4*Nfib*) compared to the *Nfib* low-expressing and KO ones (LM1 *Nfib*KO, LM9 *Nfib*KO, 4T1 *Nfib*KO and HR1gCtrl) (**New Figure 4C and Materials and Methods section**). In conclusion ERO1L is a direct transcriptional target of NFIB.

Reviewer #1: Further, it is unclear how in basal like breast cancer *NF1B* levels are accumulated. Is the gene located in a commonly amplified region? Is it frequently mutated? Are there any signaling pathways that upon miss regulation cause accumulation of *NF1B*.

Authors' response: We thank the Reviewer for these questions. *NF1B* DNA copy number is amplified in 40-55% of basal-like breast cancers (PAM50 classification) and iC10 tumours (intClust classification, Dawson et al., 2013) in both the METABRIC (Curtis, C. 2012) and TCGA breast cancer (Koboldt et al. 2012) cohorts (**New Appendix Figure S7C and D and Materials and Methods section**). *NF1B* mRNA expression is also higher in basal-like breast cancer (PAM50 classification) and in iC10 (intClust classification) compared with other breast cancer subtypes in both the METABRIC and TCGA breast cancer cohorts (Figure 6C, **New Appendix Figure S7B**). Cluster iC10 is dominated by TNBC subtype (83% of iC10 labelled tumours are TNBC (Mukherjee et al. 2018)).

NF1B is rarely mutated (somatic mutation frequency: 0.2%), and only two missense mutations were found for *NF1B* (**Figure 1 for the Reviewers**) one in an intergenic region and the other in the CCAAT box-binding transcription factor (CTF) domain.

Figure 1. Number and type of *NF1B* mutation in breast cancer. Graphical representation of *NF1B* mutations from cBioportal (Cerami et al. 2012; Gao et al. 2013).

NF1B levels can be regulated by miRNAs and Drosha (Fujita et al. 2008; Zhou et al. 2014; Becker-santos et al. 2016; Rolando et al. 2016; Rolando and Taylor 2017), and by several transcription factors (*e.g.*, ASCLZ, MYC, PAX6 and BRN2) in different cell types (Borromeo et al. 2016; Mollaoglu et al. 2017; Ninkovic et al. 2013; Fane et al. 2017), whether such regulation occurs also in breast cancer is unknown.

Reviewer #1: Another mechanistic relevant point is how *ERO1A* operates to support metastasis. As described, it is genetically relevant but mechanistically is unclear how an oxidoreductase located at the endoplasmic reticulum is functioning to support metastasis. Is *ERO1A* catalytic activity necessary? Finally, the relationship with *VEGFA* is referred in the literature but the current depicted data in the manuscript reads largely correlative and not causative.

Authors' response: We thank the Reviewer for these comments and questions. To address if *ERO1A* catalytic activity is necessary we quantify cellular ROS formation because the formation of a disulphide bond by *ERO1A* results in increased ROS levels such as hydrogen peroxide (Zito, 2015). Notably, we found that *NF1B/Nfib-ERO1A/Ero1l* overexpressing models produce more ROS than the *NF1B/Nfib-ERO1A/Ero1l* low-expressing cells (**New Figure 5A**). Moreover, it has been shown that ROS stabilises HIF1 α protein through inhibition of the prolyl hydroxylase enzyme (Bell et al., 2007; Yan et al., 2010). In the *NF1B/Nfib-ERO1A/Ero1l* overexpressing cells, we found stabilization of HIF1 α in the nucleus (**New Figure 5B**), which resulted in upregulated *VEGFA/Vegfa* mRNA (**New Figure 5C and New Appendix Figure S6A**) and protein levels (**New Appendix Figure S6B and C**

and New Figure EV4A-D), thus enhancing angiogenesis (New Figure 5D and E and New Figure EV4E-G).

Reviewer #1: *Overall, there is the perception that further development on the mechanistic part is needed to robustly explain when and how NFIB is overexpressed in breast cancer, how it mediates ERO1A expression and how the latter supports colonization.*

Authors' response: We thank the Reviewer for her/his comment. In the revised version of our manuscript, we provide additional data that show increased *NFIB* mRNA level and gene copy numbers specifically in the TNBC/basal subtype (New Figure 6 and New Appendix Figure S7). Moreover, we discovered that ROS, which are produced by ERO1A, upregulate *VEGFA/Vegfa* mRNA levels and increase VEGFA protein abundance in *NFIB/Nfib-ERO1A/Ero1l* overexpressing models via stabilisation of HIF1 α protein in the nucleus, finally promoting angiogenesis (New Figure 5, New Figure EV4 and New Appendix Figure S6).

Reviewer #1: *Several plots are based on small numbers (n=3 or n=4). I wonder how powered are these analyses and how many times repeated. Although the magnitude of the effect may seem large, this relies on 1 or 2 mice. Please comment.*

Authors' response: We thank the Reviewer for her/his comment and apologize for this lack of clarity. In the previous version of the manuscript biological replicates were reported in the figure legends, and the number of technical replicates in the methods section. In the current version we added the number of technical replicates and biological replicate to each figure legend (Figure legends and Statistical data analysis section in Materials and Methods). *In vivo* experiments were always performed with $n \geq 3$ mice per group.

Reviewer #1: *The luminescent/fluorescent mice images shown are depicted using different intensity range. Please homogenize the range across figures as otherwise is confusing. Some small lesions seem huge when displayed, whereas some big lesions happen to be displayed as weaker. This is somewhat confusing and in addition may lead to misleading interpretations. What is the criteria to define a metastasis in each experimental setting, a signal above background?*

Authors' response: We thank the Reviewer for her/his comment. We homogenised the intensity range across all the figures and repeated all quantifications. Of note, we had to change some pictures in order prevent luminescent signal from hiding the whole images. The criteria to define a metastasis is the intensity of luminescence quantified in the ROI area designed around the lungs (area designed for each mouse). The quantification was performed using the Living Image software (PerkingElmer) and the detection via the IVIS Lumina XR instrument. Three to five mice were imaged together with the same time of exposure. For the orthotopic metastasis assay, the average of the radiance of the two groups is displayed (e.g., New Figure 3B). For the experimental metastasis assay, the average of the radiance was normalized to the average of the radiance quantified on the day of cell injection (background) (e.g., New Figure 3D).

Minor points

Reviewer #1: *1-In figure 3C it is unclear what lines the p-value represents. Please address*

Authors' response: We thank the Reviewer for her/his comment, we have changed the figure accordingly (**New Figure 2D**).

Reviewer #1: *2- Some of the KM plots in the last figure are not significant. Whereas a trend in the clinics may be used in some occasions here is pointless. Indeed, this would suggest that VEGFA is not related, clinically speaking, to ERO1A and NF1B as it negatively impacts on the magnitude of the genes association with the clinical outcome.*

Authors' response: We thank the Reviewer for her/his comment. In order not to confuse the readership, we removed the old Figure 6E. We performed immunohistochemistry staining in 13 TNBC PDX models and found a correlation between NF1B, ERO1A, VEGFA co-overexpression and metastatic potential (**New Figure 6A and B and New Appendix Figure S7A**).

Point-by-point Reply to Reviewer #2:

Reviewer #2 (*Comments on Novelty/Model System for Author*):

The initial model is interesting, however the choice of cell models to follow up in down-stream experiments (or more to the point, which ones are excluded) is curious and needs further consideration. The data needs more depth of support to warrant publication in a journal such as EMBO Mol Med.

Authors' response: We thank the Reviewer for her/his comment. In the revised version we added more models for the *Ero11* validation experiments (**New Appendix Figure S5**).

Reviewer #2 (*Remarks for Author*):

The paper by Zilli et al., provides compelling evidence that NFIB promotes metastasis in breast cancer using a novel unbiased Piggy bac screen. These are important observations and is commensurate with an emerging role for NFIB as a driver of invasion and metastasis in other cancers including SCLC and melanoma. The support for NFIB as a metastatic driver is thorough and well supported, however the subsequent elucidation of the down-stream players in this metastatic/colonization axis are perhaps somewhat over-interpreted in the context of the data available. Particularly the pivotal role of ERO1A in facilitating the NFIB promotion of metastasis, particularly the data presented in figure 5 and the contention that NFIB-ERO1A induces angiogenesis. These conclusions need additional supporting data to be able to confidently draw some of the conclusions made with respect to the NFIB-ERO1A-VEGFA axis. Over-all this is a good manuscript, with important and novel observations, that I think is worthy of being published in EMBO Molecular Medicine. However, there are some issues with respect to clarity and breadth and depth of data that need to be addressed, some of which I have commented on below.

Authors' response: We thank the Reviewer for the accurate summary of our findings.

Reviewer #2 *General comments.*

1) Interestingly NFIA also appears in the Piggy Bac screen however is not really commented on with the focus falling to NFIB (not surprisingly as it is statistically significant) and FOXP1 (which isn't statistically significant. Given that NFIA is a related family member it is probably worth noting its appearance in the dataset. Is there a potential functional redundancy between family members?)

Authors' response: We thank the Reviewer for this comment and the question. We decided to exclude *NFIA* from the validation process for two reasons: First, because the enrichment of integrations in the metastatic samples is not statistically significant compared with the integrations in the primary tumours (we added all the *p* values to the respective genes in **Figure 1C**). Second, because analysis of mRNA expression levels shows that *NFIA* is not consistently overexpressed in the more metastatic cell lines (LM1, LM8 and LM9) (**Figure 2 for the Reviewers and New Figure 1F and G**). These analyses led us to the conclusion that

there is no potential functional redundancy between *NFIB* and *NFIA* in the metastatic phenotype we observed and validated.

Figure 2- *Nfia* is not consistently overexpressed in the LM metastatic cell lines. Bar graph representing mean *Nfia* mRNA expression in LM and HR1 cell lines. $n = 2$ biological replicates and $n = 2$ technical replicates, means \pm s.d., two-tailed Student's *t*-test, FC= fold change.

Reviewer #2 Some clarification is needed with respect to the cell line pools used in these studies. Were the growth characteristics of each line analysed and found to be similar? If when they are pooled are we truly seeing the effect of the oligo pool or dominance by one or two of the sub-lines?

Authors' response: We thank the Reviewer for this comment and questions. Single clones of the *Nfib* KO were not characterized. Because the parental cell lines (e.g., LM1, LM9) derived from metastases foci, we wanted to preserve this diversity in the KO models, hence we pooled them. Given that the observed phenotype is a dramatic metastases impairment, it is unlikely that only one or two of the sublines was dominant.

Reviewer #2 In figure EV4 were the g3 and g4-*NFIB* lines pooled and injected into 16 mice, or individually into mice? I suspect the former, but were these lines analysed independently at all?

Authors' response: We thank the Reviewer for this question and we agree that further clarifications are needed. g3*Nfib* and g4*Nfib* were not pooled together, we selected and used only g4*Nfib* for our experiments (**New Figure EV3A and Material and Methods section**). Different guides were designed in order to select the most efficient ones for overexpression of *Nfib*. Text, figures and figure legends have been changed accordingly.

Reviewer #2 What is the rationale in choosing cells to pool? For example EVFig 2 shows 5 lines of *LN1 kd* were used to create oligo pools but only 2 lines for the LM9. Was there a rationale behind excluding other LM9 lines?

Authors' response: We thank the Reviewer for this question and comment. We pooled the clones with the complete KO at the protein level.

Reviewer #2 In EVFig 3A clones 5 and 7 were used as a WT control however the larger NFIB isoform around 50kDa is convincingly ablated. Are these cells an appropriate control to use here? Were the parental cells used as a comparator as well?, as based on the western data in EVFig 3A they don't appear to be an appropriate control line.

Authors' response: We thank the Reviewer for this question and agree with his/her comment. As controls we used: 4T1 WT clones 5 and 7, and parental 4T1 cells (**New Appendix Figure S2A and source data Appendix Figure S2**). The apparent ablation of the larger NFIB isoform was due to the short exposure time of the immunoblot. We now report an image from a longer exposure time (**New Appendix Figure S2A**).

Reviewer #2 Specific Comments

Reviewer #2 Figure 5. Identification of a novel NFIB down-stream target in ERO1A is novel and is certainly supported by the western blot analysis in EVFig 5C across the cell lines used in the study. It is curious though that mRNA analysis for validation is then restricted to the LM1/9 k.o cells. Why not include all lines analysed at the protein level? Are the mRNA changes consistent with the protein changes in all cell models?

Authors' response: We thank the Reviewer for this question. We now provide the results of *Ero1l* mRNA levels in all cell lines and tumours showing consistency with protein abundance (**New Appendix Figure S4C and D**).

Reviewer #2 Figure 5. Similarly, it is unclear why such a crucial observation as the significant role ERO1A plays in NFIB driven metastasis is only followed up in the SUM159PT cell line. Given that the data contends that the promotion of metastasis by NFIB is almost exclusively reliant on the ERO1A down-stream target it would be important to confirm this relationship in more than a single cell line model, even if the gene expression data supports the control of ERO1A expression across multiple lines. This observation needs to be followed up in additional lines used here.

Authors' response: We thank the Reviewer for this comment. Downregulation of *Ero1l* in additional models (LM1 and HR1 g4Nfib cells) using two doxycycline inducible shRNA constructs (**New Appendix Figure S5A**), reduced metastatic colonization after tail-vein injection (**New Appendix Figure S5B-E**).

Reviewer #2 Figure 5. I think the link with angiogenesis is interesting and can be suggested from some of the data but I feel the data lacks depth and strength to draw a strong conclusion that NFIB-ERO1A induces angiogenesis, as the section title in the results states and the discussion affirms. The VEGFA link is largely drawn from the previous reports by Tanaka, 2016, but the results state that "VEGFA expression increased in cells over expressing NFIB and decreased in cell lines and lung mets upon ERO1A down-regulation". However, again data is only shown here for the SUM159PT line, the analysis only confirms mRNA changes, there is no evidence that protein levels change, no VEGFA staining in the tumor sections.

Was VEGFA expression altered in the RNA-seq analysis that identifies ERO1A downstream of NFIB? Was VEGFA expression altered in all of the cell line models reported here at both RNA and protein levels?

Authors' response: We thank the Reviewer for this comment and questions. ELISA revealed increased VEGFA levels in the supernatant of *Nfib/NFIB* overexpressing cells (**New Appendix Figure S6B and C**). Consistently, IHC of tumours overexpressing *Nfib* showed increased VEGFA (**New Figure EV4A and B**). Moreover, VEGFA protein was observed in lung metastases from mice injected with SUM159PT *gNFIB*, HR1 *g4Nfib* and LM1 *shCtrl* (**New Figure EV4C and D**).

Yes, *Vegfa* is upregulated in both RNA-seq derived from tumourspheres and tumours ($p = 7.31e^{-10}$ and $p = 0.057$ respectively, **highlighted in Dataset EV2 and 3**). Of note, the tumours contain additional cell types which probably accounts for the difference in the p -values.

We now provide the results of *Vegfa* mRNA levels in all models (**New Appendix Figure S6A**) and show its upregulation in *Nfib* overexpressing cells.

Reviewer #2 *Figure 5. The interpretation of NFIB-ERO1A playing a role in angiogenesis needs more support than the IHC of lung metastases shown in Figure 5E. Were primary tumor samples analysed for CD31 staining? Additional angiogenesis data is needed and ideally using an alternative angiogenesis assay in addition to the inferences drawn from tumour sectioning shown here. Will conditioned media from NFIB over-expression (and related eg ERO1A k.d) models modulate endothelial tube formation? Is VEGFA truly the culprit here if this is the case? What about VEGFC?*

Authors' response: We thank the Reviewer for this comment and questions. IHC revealed higher CD31 expression in *NFIB/Nfib* overexpressing tumours compared with the low-expressing ones (**New Figure EV4E and F**).

To evaluate angiogenesis induction, we performed a matrigel-based tube formation assay. We found that conditioned medium from *NFIB/Nfib/Ero1/ERO1A* overexpressing cells increased endothelial tube formation compared with the low-expressing ones (**New Figure 5E**). Moreover, recombinant human VEGF₁₂₁ added to the conditioned medium from *ERO1A* low-expressing lines increased tube formation (**New Figure 5E**).

VEGFC has not been chosen because it is not upregulated in both RNA-seq (**Highlighted in Dataset EV2 and 3**).

Reviewer #2 *Figure 6: Much of this data is drawn from bio-informatic analysis of RNA based data sets and provides some interesting correlations. Despite compelling supportive evidence from the cell line models, it is important to confirm this correlation at the protein level using human tumor samples. Is there a correlation between expression of NFIB, ERO1A and VEGFA in breast cancer tumor sections? Is there significant heterogeneity in breast cancer tumors whereby a subset with a propensity to metastasis are marked by elevated NFIB and down-stream targets highlighted in the manuscript?*

Authors' response: We thank the Reviewer for this comment and questions. *NFIB* is mainly upregulated, amplified, and a poor prognosis factor in the TNBC subtype (**New Figure 6 and**

New Appendix Figure S7). Therefore, we performed IHC using our collection of TNBC PDX models. We found that the NFIB abundance is sufficient to discriminate metastatic *versus* non-metastatic PDX models (**New Figure 6A and B and Appendix Figure S7A**). Interestingly, we found a correlation between NFIB, ERO1A and VEGFA co-expression and the metastatic potential in the PDX models.

Point-by-point Reply to Reviewer #3 :

Reviewer #3: *Adequate model systems were used in the manuscript including both experimental and spontaneous metastasis models to validate the role of NFID and ERO1A in breast cancer metastasis.*

Authors' response: We thank the Reviewer for highlighting our model systems used for the validation.

Reviewer #3 (Remarks for Author): *In this study by Zilli et al, the authors take a comprehensive in vivo approach (transposon insertional mutagenesis system) to screen a unique factor that promotes tumor metastasis and identifies a transcription factor nuclear factor IM (NFIB) is enhanced in highly metastatic murine mammary cancer clones. The importance of NFIB in tumor growth and metastasis was validated by gain- and loss-of-function studies in mice. The authors claim that NFIB contributes to metastasis by managing gene level of an oxidoreductase ERO1A, that manages VEGFA gene expression that contributes to angiogenesis and consequently promotes metastasis in mammary cancer. The authors also validates the clinical relevance of their finding in patients with triple negative breast cancer by analyzing publicly available database. Overall, this study has shown that NFIB, ERO1A, and VEGFA are critical factors for tumor progression in breast cancer, which is interesting and relevant to the cancer community. However, several issues should be addressed through additional experiments and clarification to provide firmer mechanistic evidence in support of the major conclusions.*

Authors' response: We thank the Reviewer for this accurate summary and for pointing to the important messages and significant implications of our work.

Reviewer #3: *1. Throughout the manuscript, the authors claim that the effect of NFID on tumor progression is mediated by increased expression of ERO1A. Authors also suggest that the NFID-ERO1A-VEGFA signaling axis is a metastasis driver. However, additional data are needed to strengthen this claim. ERO1A is chosen as a transcription target of NFID based on previous study done in different settings and the current data only shows that knockout of ERO1A decrease metastasis in vivo. This does not demonstrate that ERO1A is a significant downstream effector of NFID or a direct transcription target of NFID. The authors need to overexpress ERO1A in NFID ablated cells and see whether the phenotype lost by NFID knockout is rescued by ERO1A. In addition to data demonstrating that NFID knockout results in ERO1A expression decrease, the authors should provide additional evidence that NFID is a transcription factor of ERO1A in mammary cancer cells, for instance through ChIP and promoter activity assays in mHR1 or 4T1 cells.*

Authors' response: We thank the Reviewer for these comments. To address how NFIB induces ERO1A expression, we performed chromatin immunoprecipitation (ChIP) for Nfib in LM1/LM1 *Nfib*KO, LM9/LM9 *Nfib*KO, 4T1 WT, 4T1 *Nfib* KO, HR1gCtrl and HR1g4*Nfib* and found that Ero11 is a direct target of Nfib. Particularly, we extracted the sequence of *Ero11* peaks from published ChIP-seq data of *Nfib* in the mammary gland (Shin et al., 2016, acc: GSE42900) using Cistrome Data Browser (<http://cistrome.org/db>), Integrated Viewer

Genome software (J. T. Robinson et al. 2011) software and designed primers on the *Ero1l* promoter region peaks. Notably, qPCR analysis revealed increased NFIB recruitment in all the *Nfib* overexpressing models (LM1, LM9, 4T1 WT and HR1g4*Nfib*) compared with the low-expressing ones (**New Figure 4C and Materials and Methods section**). In conclusion, ERO1L is a direct transcriptional target of NFIB.

Furthermore, we overexpressed *Ero1l* in the *Nfib* ablated cells (**New Figure EV5A**) and found increased metastatic colonization after tail-vein injection (**New Figure EV5B and C**).

Reviewer #3: 2. *On a related note, connection between NFIB-ERO1A and VEGFA is not clear. The authors show that ERO1A knockout results in decreased gene expression of VEGFA. To clearly state that VEGFA is the last effector molecule in the NFIB-ERO1A-VEGFA signaling axis, the authors should additionally knockout NFIB and show the decreased gene and protein expression of VEGFA. It is also worth to know whether the overexpression of ERO1A in cells with NFIB loss, not only results in enhanced metastasis potential but also restored VEGFA expression. In addition, the authors should at least discuss how ERO1A governs gene expression of VEGFA.*

Authors' response: We thank the Reviewer for these comments. We downregulated *Nfib* in HR1 g4*Nfib* cells using 3 different siRNAs and observed a decrease in *Vegfa* mRNA and VEGFA protein levels (**New Appendix Figure S6D and E**).

Moreover, IHC of lung metastases from mice injected with *Ero1l* overexpressing cells showed restored *Vegfa* mRNA and VEGFA protein levels (**New Figure EV5D and E**).

Finally, we found that ERO1A-evoked increase ROS levels stabilizes HIF1 α , upregulates VEGFA levels and enhances angiogenesis (**New Figure 5 and New Figure EV4 and New Appendix Figure S6**).

Reviewer #3: 3. *NFIB has been implicated in tumor metastasis of breast cancer in previous studies and a proposed mechanism is through suppression of CDKN2A (PMID 30350349). Whereas, the authors suggest that NFIB works through ERO1A in breast cancer progression. It is worth to know whether which factor serves as a predominant downstream effector of NFIB or whether they coordinately contribute to NFIB mediated breast cancer metastasis.*

Authors' response: We thank the Reviewer for this comment. In the cited paper (Liu et al. 2019), NFIB promotes tumour formation through the suppression of CDKN1A. RNA-seq of tumourspheres and primary tumours derived from *Nfib* high and low-expression cells show no difference in *Cdkn1a* or *Cdkn2a* expression (**highlighted in Tables EV2 and 3**).

Reviewer #3: 4. *In line with the comment above, the data in current manuscript are all originated from murine cell lines including LB-mHR1 and 4T1. The factors identified could be specific in murine models. The authors should validate the role of these key factors in human breast cancer cells.*

Authors' response: We thank the Reviewer for this comment. We already validated the NFIB metastatic axis in SUM159PT, a human TNBC cell line (**New Figure 3 and New Figure EV3**). Specifically, overexpression of NFIB from its endogenous promoter (**New Figure EV3C**) increased tumour and metastasis formation, and shortened survival of the

animals (**New Figure 3E and F and New Figure EV3D-F**). Interestingly, we found a correlation between NFIB, ERO1A and VEGFA co-expression and the metastatic potential in the PDX models (**New Figure 6A and B**).

Reviewer #3: 5. *The importance of NFID in tumor growth is demonstrated in Figure 2. The role of NFID in tumor metastasis is shown in Figures 3 and 4. The study focuses on validating NFID level and function using highly metastatic LM1,9, and 8 clones. It is unclear whether the main contribution of NFID is through promoting tumor proliferation or angiogenesis, or even through migration and invasion as others suggest in melanoma (PMID: 28119061). At least, in vitro studies validating and comparing the effect of NFID modulation on proliferation, angiogenesis, migration, and invasion using mHR1 cells are needed.*

Authors' response: We thank the Reviewer for this comment. To address this point, we performed several *in vitro* and *in vivo* studies and analysis (**New Appendix Figure S3**). *Nfib* overexpression enhanced proliferation *in vitro* (**New Appendix Figure S3A**) and also tumour onset (**New Figure EV3B**). IHC revealed increased CD31-positive blood vessels in lung metastases from mice injected with HR1g4*Nfib* cells (**New Appendix Figure S3E**). Furthermore, *Vegfa* levels were upregulated in *Nfib* overexpressing cells and tumours (**New Figure EV4G and New Appendix Figure S6A and C**). Finally, NFIB overexpression increased tumourspheres formation, migration and invasion *in vitro* (**New Appendix Figure S3B-D**). In conclusion, the data show that NFIB contributes to several steps of the metastatic cascade.

Reviewer #3: 6. *Figure 6 provides clinical relevance demonstrating that NFID or/and ERO1A and VEGFA levels correlate with worse clinical outcome. Although these data provide evidence that each factor are linked with patient survival, the connections between NFID, ERO1A, and VEGFA are not proven. Correlation study linking each factor to one another is needed. In addition, the analyzed data is limited to triple negative breast cancer patients. It is worth to know whether the NFID signaling is unique for breast cancer or it is a common critical tumor promoting factor for other solid tumor types.*

Authors' response: We thank the Reviewer for this comment and questions. NFIB is mainly upregulated, amplified, and a poor prognosis factor in the TNBC subtype (**New Figure 6 and New Appendix Figure S7**). Therefore, we performed IHC using a collection of TNBC PDX models. We found that the NFIB abundance is sufficient to discriminate metastatic *versus* non-metastatic PDX models. Interestingly, we found a correlation between NFIB, ERO1A and VEGFA co-expression and the metastatic potential in the PDX models (**New Figure 6A and B; Appendix Figure S7A**).

Previous studies showed that NFIB governs epithelial melanocyte stem cell behaviour and facilitates melanoma cell migration and invasion (Chang et al., 2013; Fane et al., 2017). NFIB has also been implicated in models of small cell lung carcinoma (SCLC) and breast cancer (Denny et al., 2016; Semenova et al., 2016; Campbell et al., 2018; Liu et al., 2019; Moon et al., 2011), and NFIB has been associated with different tumour types (Becker-Santos et al., 2017; Yang et al., 2018; Zhou et al., 2017) (**Please see Discussion part**).

Reviewer #3 #3: *Minor suggestions for expanded view (EV) figures are listed below.*

EV 1C-1D: Tumor volume and latency of 9 groups are shown in each graph which are difficult to distinguish. The tumor volume and latency should be additionally shown as bar graphs as shown in EV Figure 1E for metastasis potential.

Authors' response: We thank the Reviewer for this comment. We modified the figure accordingly (**New Figure EV1D and E**).

Reviewer #3: *EV 4A: NFIB expression in two NFIB knock-in clones (g3 and g4) are shown. Lanes 2-3 are g3 clones and lanes 4-5 are g4 clones. However, the expression level of NFIB differ between lanes (lane 2 vs 3, lane 4 vs 5). The authors need to clarify.*

Authors' response: We thank the Reviewer for this comment. To achieve NFIB overexpression cells were cultured for two months (**New Figure EV 3A legend and Methods section**). We modified the figure legend and methods section accordingly.

Reviewer #3: *EV 5: This should be moved to main figures as it is a critical data showing the connection between NFIB and ERO1A.*

Authors' response: We thank the Reviewer and have changed the figures accordingly (**New Figure 4A and B**).

12th Jan 2021

Thank you for the submission of your revised manuscript to EMBO Molecular Medicine, and please accept my apologies for the delay in getting back to you, which is due to the limited staff and increased submitted manuscripts during the holiday season. We have now received the enclosed reports from the 3 referees who re-reviewed your manuscript. As you will see, they are supportive of publication, and we will be able to accept your manuscript pending the following final minor amendments:

1) Main manuscript text:

- Please answer/correct the changes suggested by our data editors in the main manuscript file (in track changes mode). This file will be sent to you in the next couple of days. Please use this file for any further modification.
- Material and methods: Please indicate the origin of the mice. Please provide the dilutions for primary antibodies (immunoblotting / immunofluorescence).
- Please replace "Disclosure of potential conflicts of interest" by "Conflicts of interest"
- References: please list 10 authors before et al.
- Please indicate in the figures or in the legends the exact p= values, including for non-significant (n.s.) p values. Some people found that to keep the figures clear, providing a supplemental table with all exact p-values was preferable. You are welcome to do this if you want to.
- Thank you for providing a data availability section. Please note that the data have to be made publicly available before acceptance of the manuscript.

2) Figures:

- Please rename the EV tables "Dataset EV1" etc.
- Thank you for providing Source Data. Could you please upload them so as to have 1 PDF file per main figure, and 1 file for EV and Appendix figures?

3) Thank you for providing a synopsis. I included minor edits to match our style and format, please let me know if you agree with the following, or amend as you see fit:

Transcriptional factor nuclear factor IB (NFIB) is sufficient to enhance lung metastatic colonization via enhanced angiogenesis, thus revealing a targetable network that promotes breast cancer colonization.

- NFIB was identified via an unbiased ex vivo piggyBac (PB) transposon insertional mutagenesis screen, and validated as an inducer of metastatic colonization in breast cancer.
- NFIB directly enhances ERO1A oxidoreductase expression, which in turn increases intracellular ROS levels, stabilizes HIF1 α protein in the nucleus and upregulates VEGFA expression.
- Functionally, the NFIB-ERO1A-VEGFA axis enhances angiogenesis, promotes metastatic colonization and shortens overall survival of the animals.
- A correlation was found between NFIB, ERO1A, and VEGFA co-expression and the metastatic potential in PDX models.

4) As part of the EMBO Publications transparent editorial process initiative (see our Editorial at <http://embomolmed.embopress.org/content/2/9/329>), EMBO Molecular Medicine will publish online a

Review Process File (RPF) to accompany accepted manuscripts.

In the event of acceptance, this file will be published in conjunction with your paper and will include the anonymous referee reports, your point-by-point response and all pertinent correspondence relating to the manuscript. Let us know whether you agree with the publication of the RPF and as here, if you want to remove or not any figures from it prior to publication.

I look forward to receiving your revised manuscript!

With my best wishes,

Lise

Lise Roth, PhD
Editor
EMBO Molecular Medicine

To submit your manuscript, please follow this link:

Link Not Available

The system will prompt you to fill in your funding and payment information. This will allow Wiley to send you a quote for the article processing charge (APC) in case of acceptance. This quote takes into account any reduction or fee waivers that you may be eligible for. Authors do not need to pay any fees before their manuscript is accepted and transferred to our publisher.

***** Reviewer's comments *****

Referee #1 (Comments on Novelty/Model System for Author):

This is a revised version of the manuscript. Several requests were made by the reviewers to upgrade and clarify the models used. Definitely, the manuscript has largely improved through the process.

Referee #1 (Remarks for Author):

The manuscript has substantially benefited from the revision process. In particular, the authors have address all my comments providing new experimental data to support or clarify the previously made interpretations. In this regard, the mechanistic part of the manuscript related to NFIB and its direct action to induce ERO1A expression is settled. Similarly, explanations have clarified how NFIB is accumulated in basal-like tumours and how ERO1A operates to support VEGFA expression. Finally, the clinical data is now strongly backed up by IHC analyses and the clinical correlations centred to the relevant questions.

In summary, this reviewer is satisfied and feels the manuscript is substantially improved.

Referee #2 (Remarks for Author):

I commend the authors on their efforts to address the reviewers comments and clarification of key points. The revised manuscript is significantly improved and I am happy to recommend publication in EMBO Molecular Medicine.

Referee #3 (Comments on Novelty/Model System for Author):

Adequate model systems were used in the manuscript including both experimental and spontaneous metastasis models to validate the role of NFID and ERO1A in breast cancer metastasis.

Referee #3 (Remarks for Author):

The authors have adequately addressed all the concerns raised during the initial review.

The authors performed the requested editorial changes.

YOU MUST COMPLETE ALL CELLS WITH A PINK BACKGROUND ↓
PLEASE NOTE THAT THIS CHECKLIST WILL BE PUBLISHED ALONGSIDE YOUR PAPER

Corresponding Author Name: Mohamed Bentires-Aij
Journal Submitted to: EMBO Molecular Medicine
Manuscript Number: EMM-2020-13162